# Molecular exploration of fossil eggshell uncovers hidden lineage of giant extinct bird

Alicia Grealy [1,2] ✉, Gifford H. Miller [3] ✉, Matthew J. Phillips[4], Simon J. Clarke [5], Marilyn Fogel[6], Diana Patalwala [7,8], Paul Rigby[7], Alysia Hubbard[7], Beatrice Demarchi[9], Matthew Collins[10,11], Meaghan Mackie [10,12], Jorune Sakalauskaite[9,10], Josefin Stiller [13], Julia A. Clarke[14], Lucas J. Legendre [14], Kristina Douglass [15], James Hansford [16,17,18], James Haile [19] & Michael Bunce[1]

The systematics of Madagascar's extinct elephant birds remains controversial due to large gaps in the fossil record and poor biomolecular preservation of skeletal specimens. Here, a molecular analysis of 1000-year-old fossil eggshells provides the first description of elephant bird phylogeography and offers insight into the ecology and evolution of these flightless giants. Mitochondrial genomes from across Madagascar reveal genetic variation that is correlated with eggshell morphology, stable isotope composition, and geographic distribution. The elephant bird crown is dated to ca. 30 Mya, when Madagascar is estimated to have become less arid as it moved northward. High levels of between-clade genetic variation support reclassifying *Mullerornis* into a separate family. Low levels of within-clade genetic variation suggest there were only two elephant bird genera existing in southern Madagascar during the Holocene. However, we find an eggshell collection from Madagascar's far north that represents a unique lineage of *Aepyornis*. Furthermore, divergence within *Aepyornis* coincides with the aridification of Madagascar during the early Pleistocene ca. 1.5 Ma, and is consistent with the fragmentation of populations in the highlands driving diversification and the evolution of extreme gigantism over shorts timescales. We advocate for a revision of their taxonomy that integrates palaeogenomic and palaeoecological perspectives.

The elephant birds of Madagascar (Aves: Aepyornithidae) were large, flightless ratites that became extinct around a millennium ago. The relatedness of elephant birds to other birds remained a mystery until several genetic studies discovered that they are sister to New Zealand's kiwi[1–3], revolutionising our understanding of avian diversification. However, the biodiversity and evolutionary relationships within elephant birds have been uncertain and unstable since they were first described over 150 years ago[4], as most species are known only from few incomplete Pleistocene-Holocene post-cranial skeletal remains from south and central Madagascar[5–7] (Fig. 1a and Supplementary Data 1).

About eight species of elephant birds across two genera were generally accepted based on morphological comparison of skeletal fossils[4] (Fig. 1c), but a recent morphometric re-evaluation of skeletal material[6,7] reclassified elephant birds into four species across three genera (*Aepyornis*, *Mullerornis* and a new genus, *Vorombe*). However, this revision remains questionable: homoplasy in morphological characters that has arisen via convergent evolution means post-cranial skeletal morphology poorly distinguishes species limits within extinct ratite taxa[8] as well as the evolutionary relationships between them. Alternatively, the use of ancient DNA (aDNA) has proven to be highly

**a** / **b**

**c** The most recent taxonomic hypotheses for elephant birds.

| Original description | | | Revised[36] | | Revised[6,7] | | Revised (this paper) | | |
|---|---|---|---|---|---|---|---|---|---|
| Family | Genus | Species | Genus | Species | Genus | Species | Family | Genus | Species |
| Aepyornithidae[93] | ‡Aepyornis[39] | hildebrandti[96] | Aepyornis[39] | hildebrandti[96] | Aepyornis[39] | hildebrandti[96] | Aepyornithidae[93] | Aepyornis[39] | hildebrandti[96] |
| | | mulleri[94] | | | | | | | |
| | | gracilis[97] | | gracilis[97] | | | | | |
| | | lentus[94] | | | | | | | |
| | | grandidieri[98*] | | medius[99] | | | | | |
| | | medius[99] | | | | maximus[39] | | | maximus[39] |
| | | cursor[94] | | | | | | | |
| | | maximus[39] | | | | | | | |
| | | titan[40] | | maximus[39] | | | | | |
| | | ingens[94] | | | Vorombe[6,7] | titan[40] | | | |
| | | modestus[99] | | | | | | | |
| | †Mullerornis[94] | betsilei[94] | Mullerornis[94] | betsilei[94] | Mullerornis[94] | modestus[99] | Mullerornithidae[20] | Mullerornis[94] | modestus[99] |
| | | agilis[94] | | agilis[94] | | | | | |
| | | grandis[20] | | Not included | | | | | |
| | | rudis[94] | | rudis[94] | | | | | |
| | Flacourtia[95] | rudis[94] | | | | | | | |

successful in the delineation of extinct bird species boundaries, phylogenetic relationships, and geographic ranges[8–12], and corroboration of elephant bird systematics by molecular methods is long overdue. Although the warm, humid environment of Madagascar is suboptimal for the preservation of aDNA in bone[13], it has been retrieved from elephant bird eggshell[3,14], which is found in abundance while skeletal fossils

are less common[15]. Aided by eggshell micro-morphology, stable isotope geochemistry and palaeoproteomics, here, we detail the first phylogeographic survey of elephant birds using eggshell whole mitochondrial aDNA, in order to revisit elephant bird taxonomy and evolutionary history. As an island with high levels of endemism, Madagascar is a model system for studying the mechanisms underlying evolution and

**Fig. 1 | Collection and characterisation of elephant bird eggshell morphotypes.**
**a** Map of Madagascar depicting the geographic location of eggshell samples collected (small circles) and analysed (larger circles with a border). Samples with genetic data are represented by their ID# and the thickness of the sample is proportional to the icon represented to its right. The location of fossil specimens of aepyornithids (diamonds) and mullerornithids (squares) are shown (Fig. 1c; see Supplementary Data 1 for locality data and references). Specimens for which DNA data were available are coloured yellow, including the four previously published genomes retrieved from bone specimens. Superscripts beside specimens refer to the literature that previously published genetic data for these specimens. Simplified topography of the landscape is shown with rivers represented by fine lines (adapted from https://commons.wikimedia.org/wiki/File:Madagascar_rivers.svg under CC BY-SA 3.0, https://creativecommons.org/licenses/by-sa/3.0/deed.en; river names omitted) and biomes represented by shades of grey (adapted from Brown et al.[93] under CC BY 4.0). **b** The distribution of eggshell thicknesses derived from the total number of eggshells collected across the north and south of Madagascar. Width of eggshell silhouettes are scaled to represent the mean thickness for the morphotype, and are positioned over the X-axis at the mean thickness; the width of the coloured bars depict two standard deviations either side of the mean. **c** Taxonomic revisions for elephant birds with superscripts cross-referencing the original author of the taxonomic name[94–100]. Source data for this figure can be found in Supplementary Data 1–4.

## Results and discussion

Over 960 elephant bird eggshell fragments were collected from 291 localities across southern, central, and, for the first time, northern Madagascar (Fig. 1a and Supplementary Data 2). 21 new radiocarbon dates show that the distribution of eggshells sampled is temporally constrained between 1290 and (at least) 6190 years BP (Supplementary Data 3), and is contemporaneous with most previously dated bone specimens from these areas[16] (Supplementary Data 1). In addition, eggshell deposits are found nearby skeletal deposits (Fig. 1a, Supplementary Data 1), indicating that the eggshell are likely associated with the same taxa that have been described from skeletal material sourced from the same geographic areas. The youngest sample dates to 1290 ± 15 years BP, suggesting that elephant birds were extant at this time but may have become extinct within the following few hundred years, which is consistent with other estimates[17,18]. Measurements of eggshell thickness reveals three morphotypes: in the south, a bimodal distribution of thicknesses is observed, with each mode corresponding to separate morphotypes of eggshell—one that is, on average, less than 1.1 mm thick, and one that is over three times as thick, exhibiting a mean thickness of 3.32 mm (Fig. 1b). Eggshells from northern Madagascar represent hitherto uncharacterised fossils with an average thickness intermediate between the morphotypes of the south at 1.95 mm (Fig. 1b). Two eggshell fragments from central Madagascar are likewise intermediate in thickness. Using phylogenetically-corrected regressions between eggshell thickness and egg mass, and eggshell thickness and bird mass from 65 birds (Supplementary Note 9), we estimate that, in life, the mass of the thinnest eggs would have been on average 0.86 kg ($\sigma = 0.24$ kg), laid by an emu-sized bird weighing ~41 kg ($\sigma = 14.83$ kg). The thickest eggs are estimated to have been an order of magnitude heavier at 10.47 kg ($\sigma = 3.16$ kg), and were laid by a bird weighing ~1000 kg ($\sigma = 413.53$ kg). Eggs of intermediate thickness weighed on average 3.18 kg ($\sigma = 1.01$ kg), and were laid by a bird weighing ca. 230 kg ($\sigma = 91.25$ kg; Supplementary Data 2).

Using hybridisation enrichment and high-throughput sequencing of aDNA extracted from eggshells of each morphotype from across these regions, we retrieved 17 near-complete (more than 14,000 bp, average coverage 27X) and four partial (more than 8500 bp, average coverage: 3X) elephant bird mitochondrial genomes (Supplementary Data 4). These genomes, as well as four previously published elephant bird genomes derived from skeletal specimens[1,2], were used to infer the phylogenetic relationships among eggshell and skeletal morphotypes (Fig. 2a). We find that mitochondrial haplotypes derived from eggshell and skeletal specimens cluster into four well-defined clades that correspond to eggshell thickness and geographical region. With support from eggshell micromorphology, protein and stable isotopes, these data reveal additional insights into elephant bird ecology and speciation, contributing to our understanding of how these birds fit into the rich evolutionary history of Madagascar.

### The elephant bird crown coincides with a shifting environment and represents family-level divergence

The first divergence within the elephant bird lineage separates all eggshell specimens thinner than 1.5 mm into a monophyletic cluster with published *Mullerornis* genomes[1,2] from bone, and all eggshell greater than 1.5 mm thick into a monophyletic cluster with published *Aepyornis hildebrandti, A. maximus* and *Vorombe titan* genomes[1,2] (Fig. 2a). These groupings receive the highest statistical support from both maximum likelihood and Bayesian approaches (Fig. 2a), and confirm that the thin eggshell morphotype is associated with the gracile genus *Mullerornis*. The average Kimura 2-Parameter genetic distance between these two clades across an ~600 bp barcode region of *cytochrome oxidase I* (*COI*) is more than ten times greater (11.9%) than the average genetic distance within each clade (Supplementary Fig. 2 and Supplementary Data 5). The high level of divergence between these clades (more than triple the average between-genus within-family genetic distance of other ratites at 3.7%, and more than double the average genetic distance between moa families) (Supplementary Fig. 2) strongly suggests they belong to more than one family, although they are currently both classified as Aepyornithidae. Analogous to the genetic distance between emu (family Dromaiidae[19]) and cassowary (family Casuariidae) at 12.8%, we advocate that *Mullerornis* be placed into a different, monogeneric family called "Mullerornithidae" (as originally advocated by Charles Lamberton in 1934[20]).

While proteins typically provide less phylogenetic signal than ancient DNA, we nevertheless sequenced the ancient proteins preserved in the eggshell matrix to explore amino acid differences. Partial sequences of putative nuclear-encoded type-1 and type-2 C-lectin eggshell proteins[21] (XCA-1, XCA-2) reconstructed from 12 eggshell specimens (Supplementary Data 7) support a family-level distinction between mullerornithids and aepyornithids, with all putatively mullerornithid eggshell (<1.5 mm thickness) having a histidine at residue 74 of the XCA-1 alignment, and all aepyornithid eggshell (>1.5 mm thickness) having a tyrosine at this site (Supplementary Figs. 4–5). Additionally, variability at positions 62 (G, A) and 65 (E, D) was observed, although more weakly supported by the raw tandem mass spectrometry data (Supplementary Fig. 6). Mullerornithids and aepyornithids appear to be missing residues found at positions 26 and 101 in all other palaeognaths, supporting their sister relationship. XCA-2 elephant bird sequences (Supplementary Figs. 5–6) also support the separation between aepyornithids and mullerornithids, with two amino acid differences (123: A, T; 130: P, T). The structure of elephant bird XCA-1 and XCA-2 match closely the structure of ostrich struthiocalcin-1 and struthiocalcin-2, respectively (Supplementary Fig. 7). Amino acid residues mainly differ in flexible regions, and even when they do not, the secondary structure is not disrupted, suggesting there is likely no functional significance to these mutations.

To explore eggshell morphological features beyond thickness, we micro-CT scanned 20 eggshells (Supplementary Data 8) representing each thickness morphotype from each region. Differences in micro-structure were detected between but not within the two clades, with

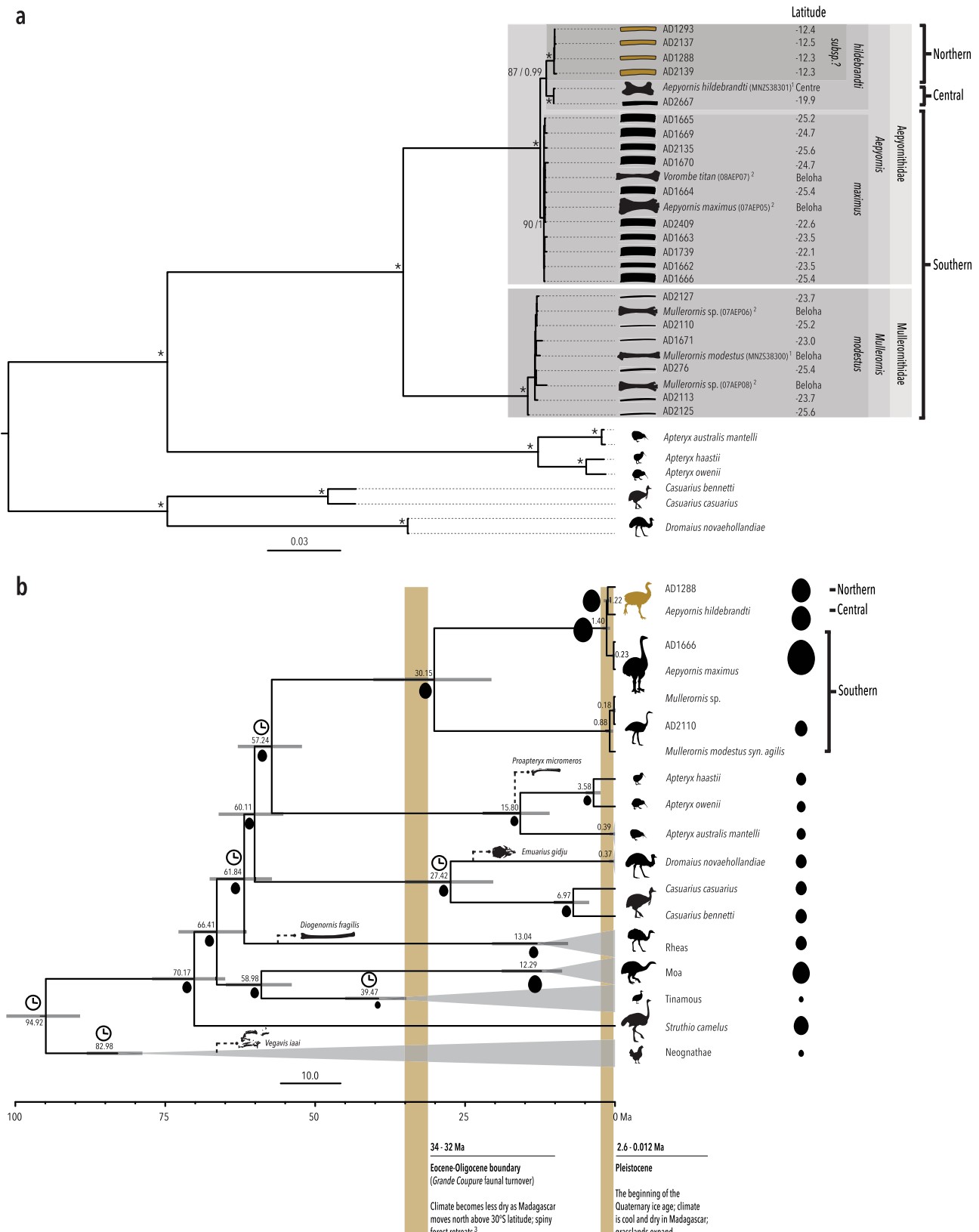

the porosity of aepyornithid eggshells being significantly higher than mullerornithid eggshells ($p = 0.032$, $df = 14$; Fig. 3 and Supplementary Table 11). This difference is driven by differences in pore density ($p = 0.031$, $df = 14$) rather than pore volume ($p = 0.198$, $df = 14$; Fig. 3 and Supplementary Table 11). Typically differences in pore characteristics cannot distinguish species but can be used to discriminate

between palaeognath orders, and families within an order[22–24], further supporting the idea the two genera belong to different families.

Molecular dating estimates that the divergence between Aepyornithidae and Mullerornithidae occurred approximately 30 Ma (95% HPD 20.6–40.3 Ma; consistent with recent studies[2,3]), close to the Eocene-Oligocene boundary, a period of marked global cooling and

**Fig. 2 | Molecular dated phylogenetic tree showing the mitochondrial relationships between eggshell specimens from northern, central and southern Madagascar. a** Consensus mitochondrial phylogenetic tree for all eggshell specimens sequenced here and genomes previously published from bone specimens. Nodes marked by an asterisk had the highest support from both ML and Bayesian analyses. Numbers beside nodes give the ML bootstrap support and Bayesian posterior probability for the topology. Unmarked nodes had ML support <70%. Superscripts beside specimens refer to the literature that previously published genetic data for these specimens. **b** Dated mitochondrial phylogeny for the palaeognathae generated in MCMC tree using several representative elephant bird taxa (see also Supplementary Fig. 3). Fossil calibrated nodes are indicated by a clock symbol, and fossil taxa used to calibrate nodes are represented by a silhouette. Grey bars represent 95% HPDs of the date estimate for the node. The timings of major geological and climatic events are indicated on the timeline below the tree. Egg mass is represented by ovoid silhouettes whose area is proportional to egg mass. Egg mass for elephant bird taxa at the tree tips was estimated using average eggshell thickness for each taxon (Supplementary Note 9). Egg mass ancestral state at internal nodes was estimated using the average eggshell thickness for each elephant bird taxon. Source data for this figure can be found in Supplementary Data 6 and 10–11. Silhouettes have been reproduced from Grealy et al.[3] with permission from Elsevier.

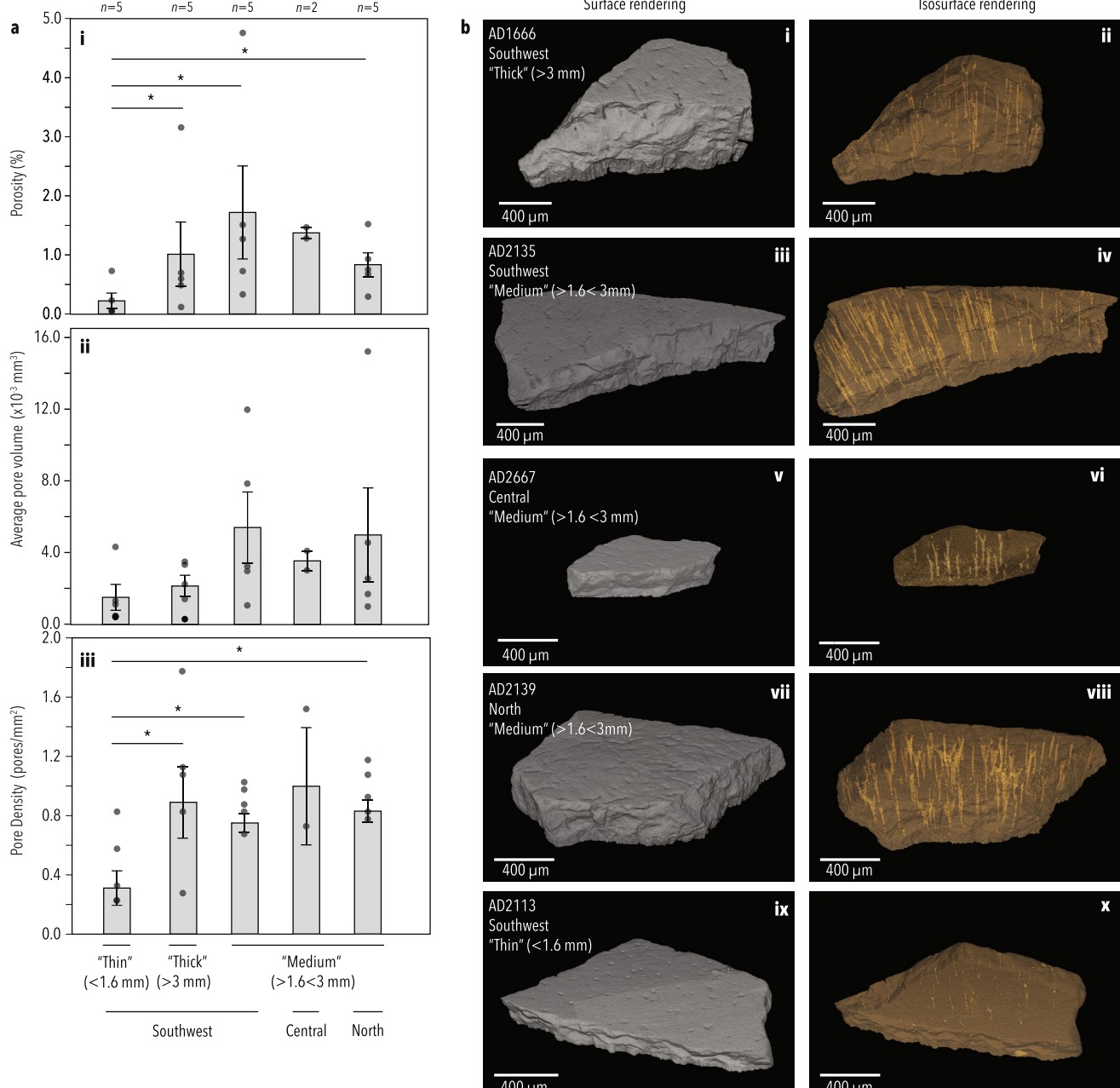

**Fig. 3 | Micro-CT characterisation of eggshell morphotypes. a** Bar chart comparing the mean porosity, mean 'average volume per pore' (proportional to pore area), and mean pore density within the ROI examined of each eggshell morphotype. Error bars represent 95% confidence intervals of the standard error. Significant differences ($p < 0.013$, two-sided Mann–Whitney $U$ test with Bonferroni correction, Supplementary Table 11) are indicated by an asterisk. **b** Representative micro-CT scans of each eggshell morphotype showing the outer surface and internal pore structures. Source data for this figure can be found in Supplementary Data 8 and 12.

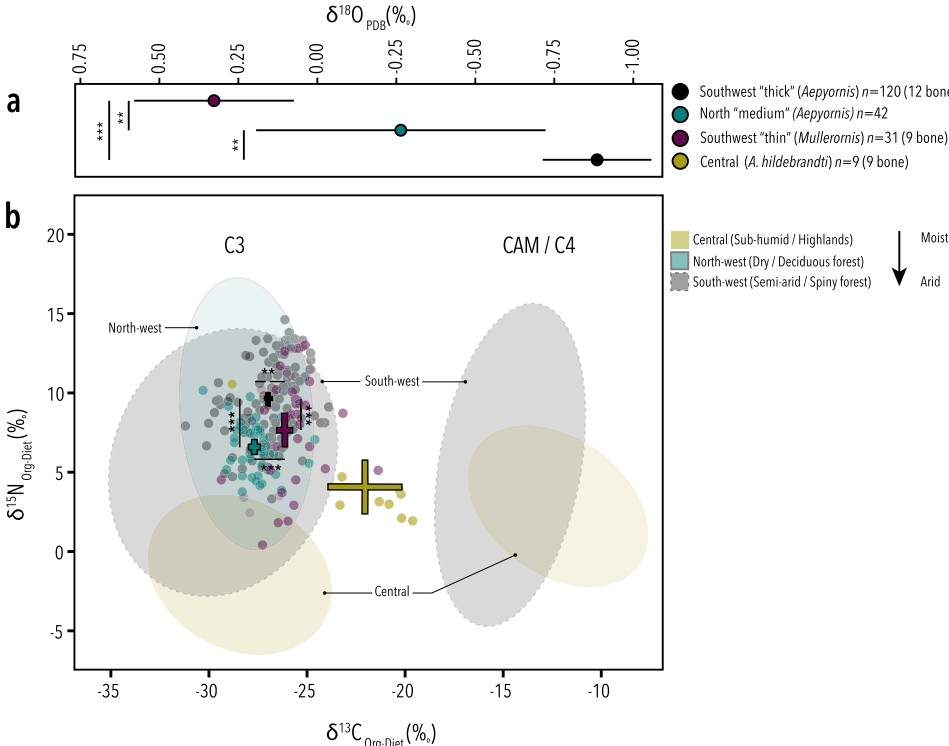

**Fig. 4 | Dietary stable isotope content of *Aepyornis* and *Mullerornis* eggshells from southern and northern Madagascar. a** The mean $\delta^{13}C_{PBD}$ isotope content. **b** The mean $\delta^{13}C_{Diet}$ and $\delta^{13}N_{Diet}$ isotope content from the organic fraction of in relation to the previously published carbon isotope distribution of C3, CAM and C4 photosynthetic plants from each bioregion (ellipses). Coloured crosses represent 95% confidence intervals of the mean for each morphotype. Asterisks indicate significant differences ($p < 0.01$, two-sided Mann–Whitney $U$ test with Bonferroni correction, Supplementary Table 12). Mean stable isotope values for all eggshell types significantly differ from the previously published isotope values in *Aepyornis hildebrandti* bone from central Madagascar. Source data for this figure can be found in Supplementary Data 9.

faunal turnover in the Northern Hemisphere. Before this time, Madagascar's climate was largely dry and the island was dominated by spiny forest; as Madagascar moved north above 30°S, and with the establishment of the circumpolar current[2], precipitation increased and the range of this biome contracted towards the southwest[25] while the humid northern and dry western forests originated and expanded[26]. The changes in palaeoclimate and dominant vegetation during this time may have driven divergence between the two families of elephant birds as has been proposed among Madagascar's lemurs[27].

Niche partitioning between sympatric aepyornithids and mullerornithids in the south is evidenced by the differences between the isotopic signatures of their eggshells. Significant differences in the stable carbon ($\delta^{13}C$), nitrogen ($\delta^{15}N$), and oxygen ($\delta^{18}O$) isotopic compositions of both the organic matter and carbonate fraction of 130 southern mullerornithid and aepyornithid eggshells ($p < 0.009$, df = 149; Fig. 4; Supplementary Figs. 9–10, Supplementary Data 9 and Supplementary Table 12) indicate that their diets differed. Although the $\delta^{13}C$ falls within the distribution of $\delta^{13}C$ for C₃-type vegetation (trees and shrubs) for both families, the mean $\delta^{13}C$ of southern aepyornithid eggshell is statistically significantly more negative than mullerornithid eggshell (Fig. 4b). An estimation of the relative contribution of CAM (Crassaculean Acid Metabolism) vegetation to elephant bird diet revealed a greater proportion in the diet of mullerornithids (22.76%) compared with sympatric aepyornithids (16.95%), consistent with other studies[28,29]. More negative $\delta^{13}C$ values in bone compared with eggshell may suggest that elephant birds bred during the dry season, as facultative CAM plants (those that switch between daytime (C₃) and night (CAM) fixation of $CO_2$) are expected to be biased toward CAM fixation during times of moisture stress[28], resulting in more positive $\delta^{13}C$ values (observed in the eggshell). A greater dependence of mullerornithids on succulents (CAM) supports the idea that they may not have relied as heavily as aepyornithids on groundwater-fed reservoirs for hydration during the breeding season[28]–a hypothesis that would account for the more positive $\delta^{18}O$ values observed in mullerornithid eggshell compared with aepyornithid eggshell (Fig. 4a). $\delta^{18}O$ values tend to be more negative in frequent drinkers compared with animals that meet water requirements through food[30]. However, the difference in $\delta^{18}O$ values between aepyornithids and mullerornithids in the south is very small (~1‰) and may instead be predicted simply by body size differences, where the larger of sympatric taxa has body water more closely reflecting the $\delta^{18}O$ of local drinking water[31]. Furthermore, the variability within taxa is also extremely small, indicating a well-buffered environmental effect (i.e., precipitation balanced with evapotranspiration): this supports the idea that drinking water sources were being constantly replenished by groundwater[28].

The mean $\delta^{15}N$ in aepyornithid eggshell from the south is 4.1‰ higher than the average of over 400 plants from that region (Supplementary Data 9), and $\delta^{15}N$ is enriched by about 1.5‰ between aepyornithid and mullerornithid eggshell from the same region. Enriched nitrogen can be an indication of nutritional stress in birds (such as during laying)[32]; however, if that were the case we might expect to see a significant difference in $\delta^{15}N$ between eggshell and bone, which we do not. Enrichment of nitrogen can also be a response to aridification and is observed in drought-tolerant herbivores living in xeric habitats[33]; however, we do not observe enriched nitrogen in mullerornithids from the same environment. Rather, the enriched $\delta^{15}N$ observed in aepyornithids most likely reflects non-steady state nitrogen metabolism associated with their giant eggs. Alternatively, southern aepyornithids may belong to a higher trophic level than

mullerornithids[34], supplementing their diet with insects or even small lizards, or, it may be further support for the nocturnal activity of aepyornithids[35], with nocturnal species having higher $\delta^{15}N$ compared with diurnal, crepuscular, or cathemeral species[30].

## Surprisingly low genetic diversity is found within elephant birds from the south

Our genetic evidence suggests that each elephant bird family is monogeneric. *Mullerornis modestus*[6] (syn. *M. agilis, M. betsilei, M. rudis, M. grandis*) skeletal specimens are consistently nested within the Mullerornithidae clade alongside all eggshell specimens <1.5 mm in thickness (Fig. 2a). Short branch lengths, recent estimated divergence, and low support for the phylogenetic topology within the clade is consistent with these samples all representing a single species. The average pairwise genetic distance in *COI* between *Mullerornis* specimens is 0.27% (±0.051%; Supplementary Fig. 2 and Supplementary Data 5), which corresponds to the average amount of genetic variation within species for all other ratites at the same locus (0.39% ± 0.431%; Supplementary Fig. 2 and Supplementary Data 5). There is also no evidence of geographic clustering within *Mullerornis* from the south, with some samples from the southwest being more closely related to samples from the far south than others from the southwest, and vice versa (Fig. 2; *p*-value = 0.135, *Z* = 0.27, Mantel's test; Supplementary Note 5). Thus, our data suggest that there was one species of *Mullerornis* inhabiting the south during the late Holocene. Coupled with recent morphometric analysis of skeletal fossils that include *Mullerornis* specimens from central Madagascar[6], our data support the notion that *Mullerornis* was a monotypic genus, having one species that was distributed throughout central/southern Madagascar, *M. modestus*; however, *Mullerornis* eggshell and bones from central Madagascar both remain to be genetically tested. Prior descriptions[36] of numerous smaller mullerornithid species in the region may have been confounded by sexual dimorphism and the lack of growth series data, as well as temporal and geographic variations in size (which has been estimated to vary by more than 50% between the late Glacial and late Holocene (over ca. 25 ka) in some moa species[37,38]).

Within Aepyornithidae, it has recently been suggested that two genera co-existed in southern and central Madagascar[6,7]—*Vorombe* and *Aepyornis*; however, we find no genetic evidence to support this hypothesis. While genetic data show that there are indeed two well-supported (90 and 87% bootstrap support) clades within Aepyornithidae, only one clade contains samples from southern Madagascar, while the other contains samples from central and northern Madagascar. The genetic distance across *CO1* between these clades is less than 1.01%: in comparison with other ratites, where the between-genus (within-family) genetic distance ranges from 2.3 to 5.1% (Supplementary Fig. 2 and Supplementary Data 5), the two aepyornithid clades are not genetically distinct enough to be considered different genera. Although there is no recognised divergence threshold for generic separation, this finding draws into question the taxonomic legitimacy of the largest ever recorded birds, *Vorombe titan*.

Furthermore, the average pairwise genetic distance in *COI* between specimens in the southern clade (0.102%; 95% CI ± 0.058%) is lower than the within-species variation of other ratites (0.39% on average; Supplementary Fig. 2), suggesting it is unlikely that more than one species, let alone genus, existed within this clade. Indeed, the two bone specimens (one identified as probable *Vorombe titan* and one as probable *Aepyornis maximus*, Supplementary Note 1) are genetically identical at this locus. There is also no correlation within this southern clade between genetic distance and geographic location (*p*-value = 0.093, *Z* = 0.31, Mantel's test; Supplementary Note 5), and no genetic difference between aepyornithid eggshell of medium thickness (1.5–3 mm) and the thickest eggshells (>3 mm) in the south, with some thinner ("medium") eggshells being more closely related to thicker eggshells than other medium eggshells, and vice versa. Branch lengths are

extremely short and the phylogenetic topology within the southern clade is not well-supported (Fig. 2), further supporting the idea there is no mitochondrial sub-structure within this clade. No differences were likewise observed in microstructure (porosity, average pore volume, and pore density; *p*-value > 0.15; Fig. 4), or isotopic signature (*p*-value = 0.1161, *F* = 2.058, *n* = 49, PERMANOVA) between medium and thick aepyornithid eggshells in the south. Amino acid substitutions were also not observed in the sequence of eggshell protein XCA-1 between any aepyornithid eggshells.

Thus, our results suggest that either we sampled only eggshell belonging to one of the two aepyornithid genera in the south, or, that one genus does not represent a valid taxonomic group. Considering that the distribution of skeletal fossils of both *Aepyornis* and *Vorombe* spatially and temporally overlap with the eggshell specimens analysed here (Supplementary Data 1), it is unlikely that we failed to sample eggshell from an entire taxon that was allegedly sympatric with the other[6], save an extreme taxonomy-dependent depositional bias (such as one genus nesting in a different region to where the greatest concentration of their skeletons can be found, or eggshell washing to the coast from elsewhere). Nevertheless, the southern aepyornithid eggshell analysed here all belong to one genus—the same genus as the central Malagasy eggshell. Because the identification of the central Malagasy skeletal specimens as *Aepyornis hildebrandti* is undisputed, the southern eggshell analysed here would also belong within the genus *Aepyornis*, but to a separate species. On this basis, we advocate tentatively synonymising *Vorombe*[6,7] with *Aepyornis*[39] and returning *titan*[40] to synonymy with *maximus*[39], pending further evidence to the contrary.

Rather than belonging to different species, the two aepyornithid skeletal morphotypes observed in the south[6] may belong to one sexually dimorphic taxon. Similar to the kiwi, the closest extant relative of elephant birds[1–3], where females can be between 120–180% the size of males[8,10], the smaller "*Aepyornis maximus*" may be males, and the larger "*Vorombe titan*" may be females of the same species[4]. Indeed, several species of moa (*Dinornis*) have been consolidated into two sexually dimorphic species[8,38], one from the North Island (*D. novaezealandiae*) and one from the South Island (*D. robustus*). On the North Island, for instance, females ("*D. giganteus*" and "*D. novaezealandiae*") were up to 280% the mass of the males ("*D. struthoides*")[8,38]. On average, *Vorombe* is 175% the size of *A. maximus*, but even considering the maximum estimated body size of *Vorombe* compared with the minimum estimated body size for *A. maximus*[6], the difference still falls within the range of (albeit extreme) reversed sexual dimorphism seen in other ratites.

This hypothesis is further supported by our calculation that the mass of the bird that laid such thick eggshell was approximately the size of *Vorombe*, whereas a bird the size of *A. maximus* would have been expected to lay a much thinner egg (see also Supplementary Note 10); though eggshell medium in thickness is found in the south, these are also genetically identical (Fig. 2) to even the thickest (>4 mm) eggshells (which may just represent inviable, unfertilised or prematurely broken eggs as eggshell becomes thinner as the embryo develops). Ultimately, sex typing is required to confirm the hypothesis that skeletal morphotypes represent within-species sexual dimorphism; however, this entails recovering nuclear DNA from bone specimens, as eggshell DNA is expected to be female (maternal origin). Alternatively, the presence of female-specific bone histology (i.e., medullary bone in gravid females) in only one skeletal morphotype may also help test the hypothesis of sexual dimorphism in Aepyornithidae; however, medullary bone has not been detected in any aepyornithid skeletal fossils thus far[41,42].

While nuclear DNA has been recovered from elephant bird eggshell, it has been limited to the enrichment of conserved protein-coding genes useful for discerning deep divergences[3]. Future advancements in DNA technology may allow highly polymorphic

regions of the nuclear genome to be recovered, revealing additional genetic diversity or population structure within elephant bird taxa that we were unable to resolve with mitochondrial DNA alone. Furthermore, sampling across more of their range (Fig. 1 and Supplementary Data 1), particularly the mid-to-north west (Besalampy), central highlands, central-east and south-east (Farafangana, Fort Dauphin) will be essential to garner the complete picture of elephant bird diversity. Regardless, these results suggest less diversity within Aepyornithidae in southern Madagascar than has been previously described. Elephant bird biodiversity was therefore not only lower than predicted from the skeletal fossil record, but also may have been much lower than one might expect from a large country with numerous climatic and geographic barriers to gene flow (Fig. 1) and one of the highest endemic biotas in the world[43]. It is possible that more species existed in southern Madagascar in the past; however, the pre-Holocene skeletal fossil record for elephant birds from this region is scant (Supplementary Data 1). Low genetic diversity may have impacted elephant birds' resilience to major changes in the environment caused by human land use in the late Holocene, contributing to their extinction.

### Evidence for hidden diversity within *Aepyornis* is recovered from eggshell in the far north

Four mitogenomes from eggshell found in Madagascar's far north places them unambiguously in a monophyletic clade sister to the central Malagasy clade that includes the published mitogenome from an *A. hildebrandti* skeletal specimen, as well as an eggshell specimen of comparable thickness to the northern eggshell (Fig. 2a). The exclusion of central Malagasy specimens from the northern clade with high confidence (and vice versa) indicates that the northern eggshells belong to a unique evolutionarily significant unit. Although the average genetic distance in *COI* between northern *Aepyornis* and central *Aepyornis* is 0.2% (within the limits of intra-specific variation observed in ratites; Supplementary Fig. 2), it is unlikely that the distinctiveness of these two reciprocally monophyletic clades can be attributed to random coalescence ($p$-value <0.05; Supplementary Note 5 and Supplementary Table 9), which may suggest the northern clade represents a cryptic taxon. On the other hand, genetic connectivity between overlapping populations of *Aepyornis* throughout Madagascar could also result in a pattern of genetic differentiation at the sampled extremes of the geographic range but what is, in reality, an effect of isolation by distance (IBD). A Mantel's test (Supplementary Note 5) shows that genetic distance is significantly correlated with geographic distance across Madagascar ($p$-value = 0.001, $Z$ = 0.31), however, this observation is also to be expected when physical obstacles to gene flow are confounded with large geographic distances. Nevertheless, IBD cannot be ruled out without additional sampling of individuals between northern and central Madagascar, presuming such populations ever existed. Another explanation could be that gene flow is maintained by male dispersal; under this scenario, mitochondrial DNA would only reflect the relationships between female elephant birds as it is inherited maternally, causing them to appear more different than the nuclear genome would.

Whether this taxon can be considered a novel species, subspecies, or merely a population of *A. hildebrandti* cannot be concluded based on these data; nevertheless, it is an independent lineage that represents novel diversity within *Aepyornis*. Prior to this finding, *A. hildebrandti* had an extremely limited geographic range with specimens restricted to only the highest elevation sites (ca. 1500 metres)[6,44]; the inclusion of the northern clade within *A. hildebrandti* would expand the known range of this species by nearly one thousand kilometres. To the best of our knowledge, no skeletal specimens of *Aepyornis* have thus far been described from the far north of Madagascar. Unlike *A. hildebrandti* (central Madagascar), whose $\delta^{13}C$ and $\delta^{15}N$ values reflect an adaptation to eating a mixture of C3 shrubs and up to 48% C4 grasses[29,44,45], the diet of northern elephant birds was dominated by C3 shrubs (Fig. 4b, $n$ = 42).

However, the average $\delta^{15}N$ is 6.6‰—about 2‰ lower than the mean of 42 plants from the northwest/dry-deciduous forest biome[46] (Fig. 4 and Supplementary Data 9), but 2.5‰ higher than *A. hildebrandti*. Two explanations for the depletion observed in dietary nitrogen could be that (a) members of the northern clade subsisted on fruits, as frugivores typically exhibit lower $\delta^{15}N$ than folivores[30], or (b) $\delta^{15}N$ is typically enriched in nocturnal species compared with diurnal or cathemeral species[29], so similar to *Mullerornis*, it may have been more active earlier in the day. This unique feeding ecology may support the designation of the northern clade as a new species; however, it may instead suggest that *A. hildebrandti* was a generalist species with the ability to subsist in a variety of environments. Though we cannot erect a formal name for this new lineage of elephant bird, the notion that an aDNA sequence, in isolation, can define a taxon has been tested with the discovery of Denisovan hominins, where the fossil record consists of few morphologically indistinct bones with significant genetic differences to modern humans[47].

The medium-thickness eggshell morphotype of the northern clade, coupled with its close relationship with *A. hildebrandti*, suggests northern elephant birds were likely smaller-bodied than their counterparts in the far-south as the bones (and eggshell) of *A. hildebrandti* are smaller than those of southern aepyornithids[4,6]. We estimate the body size of the northern taxon to be ~230 kg with an egg mass of ~3 kg based on the average thickness of the eggshell at 1.95 mm. This is similar in size to the estimated body mass of *A. hildebrandti* at 283 kg[6] (or 235 kg by our prediction), and thus, the two are both predicted to be heavier than an ostrich.

Both the central and northern clade are a different species to the southern aepyornithid clade as the average genetic distance across *COI* is 1.01% (95% CI ± 0.0526%): above the limits of within-species variation (on average 0.1%, range 0−0.45%), and within the limits of inter-specific (within-genus) variation observed in ratites and other birds[11,12] (1.02−7.5%; Supplementary Fig. 2). There is also a methionine to valine substitution at position 58 of the XCA-2 protein alignment between the southern and northern aepyornithid eggshell analysed, consistent with them belonging to different taxa. These results support the designation of the central/northern species as *A. hildebrandti* and the southern species as *A. maximus*. Interestingly, the genetic distance between *Aepyornis* species is the lowest of all ratites; as the largest birds to have ever lived, this may be a consequence of recent divergence coupled with low evolutionary rates associated with longevity, or long generation times related to large body mass[2].

### Recent divergence within *Aepyornis* is consistent with the rapid evolution of extreme gigantism

Divergence within *Aepyornis* corresponds with the onset of the Quaternary; while this was a period of glaciation in the northern hemisphere, the Malagasy climate would have been cool and dry[48]. Molecular dating estimates that there is 1.22 Ma (95% HPDs 0.6−1.9 Ma) separating the central and northern clades of *Aepyornis*: ill-adapted for eating C4 grasses (Fig. 4b), the expansion of a grassy valley known as the Mandritsara window (Fig. 1a) during the Pleistocene may have isolated montane-adapted populations of the northern highlands in forest refugia that maintained mesic conditions, while central populations of *A. hildebrandti* became adapted to an open "pseudo-steppe" habitat of the Central High Plateau[35]. Such an isolation obstacle is hypothesised to have led to the evolution of "species pairs" in other Malagasy natives[49], and indeed, the far north exhibits high levels of micro-endemism[50,51]. This timing also coincides with the diversification of endemic grass species that are adapted to grazing pressures, 1−7 Ma[52,53]. Alternatively, populations in west/north-west Madagascar (where eggshell has also been found) may have originated from those in central Madagascar expanding into lower altitude environments to follow the retreating C3 vegetation, and adapting to exploit dry deciduous forests. The split between the southern *Aepyornis* clade and

the central/northern clade occurs near-contemporaneously with the split between *A. hildebrandti* and the northern clade at 1.4 Ma (95% HPDs 0.8–2.1 Ma). Like in the north, this divergence could have been driven by a low-lying valley (Menaharaka window) separating the central and southern highlands, followed by a late expansion of *A. maximus* into the lower altitudes of the deep south that were already inhabited by mullerornithids (Fig. 1a). The extremely large body size (ca. 700–1000 kg) in *Aepyornis maximus* appears to be a derived trait, with the crown elephant birds estimated to be closer to the size of *Mullerornis* (ca. 80 kg; Supplementary Note 9 and Supplementary Fig. 12), and the common ancestor of *Aepyornis* being half that size (ca. 400–500 kg) only a mere 1.4 my ago (Supplementary Note 9 and Supplementary Fig. 12). That is, body size of birds in the lineage leading to *Aepyornis maximus* nearly doubled between the Mid- to Late Pleistocene. This is consistent with previous estimates[2], as well as the observation that most megafauna evolved large body size in recent times for more efficient thermoregulation in cooler climates[54]. Thus, the world's largest eggs are a comparatively recent development in elephant bird evolutionary history. Aridity and temperature could also have interacted on different aspects of elephant bird physiology, including eggshell morphological traits such as porosity, to drive the recent, rapid radiation within *Aepyornis*. Again, the nuclear genome of *Aepyornis* would provide further insights into the genetic control of gigantism and may identify a potential role for reproductive character displacement in the evolution of these traits.

## A revision is in taxonomy is needed that integrates palaeogenomic and palaeoecological perspectives

The systematics of elephant birds has been confused since their discovery due to the paucity of diagnostic skeletal fossils, with little additional evidence found in the last hundred years to refute or support initial classifications. An inability for morphology to delimit species where skeletal fossils are incomplete—particularly when extreme sexual dimorphism may be present—further complicates the issue. However, the excellent biomolecular preservation of fossil eggshell has provided an alternative option for an independent investigation of elephant bird systematics in parallel to skeletal morphology. To avoid taxonomic inflation, it is essential to associate eggshell and skeletal morphotypes, and the research presented here is an important leap forward in resolving the complex diversification of some of the world's largest birds. The molecular evidence derived here from fossil eggshell supports three major conclusions: (1) *Mullerornis* is sufficiently genetically distinct from all other taxa to be recognised as belonging to a separate family, "Mullerornithidae"; (2) diversity within Aepyornithidae is low (with regard to mitochondrial divergence relative to other large ratites, and compared with past taxonomic hypotheses based on skeletal morphology), with skeletal morphs potentially instead representing extreme sexual dimorphism and (3) a new eggshell collection found in Madagascar's far north is genetically distinct and represents a novel lineage of *Aepyornis* (likely *A. hildbrandti*), whose skeletal fossils await discovery pending a concerted search. We propose that a revision in elephant bird taxonomy and systematics is needed, incorporating this palaeogenomic perspective. Finally, we identified potential drivers of speciation in elephant birds, namely, the expansion of grasslands during the Pleistocene. Ancestral state reconstructions also suggest a surprisingly recent origin for extreme gigantism in aepyornithids. These findings contribute to our understanding of how elephant birds lived and functioned within Madagascar's unique ecosystems, and reinforces how aDNA from eggshell is a promising avenue for studying the evolution and extinction of terrestrial megafauna.

## Methods

No ethical approval was required to undertake this research; no humans or live animals were the subject of this research.

### Specimen collection

Local permission to carry out archaeological research was granted by the Office du Maire, Commune de Befandefa and by the Chefs de Fokontany of Andavadoaka, Nosy Ve, Antsaragnagnangy, Lamboara, Ampasilava and Salary. Permits for the export of archaeological materials for the purposes of laboratory analysis were granted by the Secretariat Général of the Ministère de l'Artisanat de la Culture et des Patrimoines, Direction Régionale de la Culture et du Patrimoine Atsimo Andrefana, Visas de Sorties Numéro 09/06- MCP/SG/DRCP.AA; Numéro 05/14-MACP/SG/DRCP.AA; Numéro 08/14- MACP/SG/ DRCP.AA in accordance with Avis Numéro 375, 02/02/1978. Permission to import fossils into Australia was granted by import permit IP15012450.

Eggshell specimens (Supplementary Note 1 and Supplementary Data 2) were collected over several field seasons from various locations in the north, south, and southwest of Madagascar (Fig. 1) and were stored at room temperature. The average thickness of each eggshell sample was calculated as the mean of the thicknesses of four sides measured with a digital caliper (Supplementary Note 2). The thickness of 197 elephant bird eggshells collected from the north, 512 eggshells randomly collected from the southwest, and 241 eggshells collected from the south of Madagascar (Fig. 1) were measured to examine the distribution of eggshell thicknesses from each region. Summary statistics for these distributions were calculated in PAST v3.11[55].

### Radiocarbon dating

Eggshell samples for radiocarbon dating (Supplementary Note 3 and Supplementary Data 3) were mechanically cleaned then reduced by 50% with the stoichiometric addition of 2N HCl *in vacuo*. Cleaned fragments were converted to graphite at the INSTAAR Laboratory for AMS Radiocarbon Preparation and Research (NSRL) before measurement by Accelerator Mass Spectrometry at the Keck Carbon Cycle AMS Laboratory at the UC Irvine (KCCAMS). Conventional radiocarbon ages have been calibrated using CALIB v.7.1 and SHcal13[56–58]. Sample AD1739 was found in an archaeological deposit by KD; eggshell from the same context were radiocarbon dated as above.

The ratio of two enantiomers (A/I), the protein amino acid L-isoleucine and the non-protein diastereomer D-alloisoleucine, was measured in eggshell by ion-exchange high-pressure liquid chromatography (Supplementary Note 4); A/I reflects time and the integrated thermal history experienced by the sample. Quality control is monitored with a laboratory standard, ILC-G[59]; 383A/I analyses of the ILC-G standard in the lab average $0.457 \pm 0.012$.

### Micro computed tomography

20 eggshell samples of varying thicknesses across each location were imaged using micro computed tomography (Skyscan 1175 micro-CT, *Bruker-microCT*) at the Centre for Microscopy, Characterisation and Analysis, The University of Western Australia (Supplementary Note 7, Supplementary Data 8); n.b., note that not every eggshell imaged yielded aDNA and some eggshell that yielded DNA did not leave enough sample to be imaged). Analyses were performed on a central slice in order to minimise the effects of imaging artifacts and potential weathering of pores close to the surfaces of the eggshell. For each sample, 2D analyses of pore density and pore area were performed on a 20.07 mm² region of interest (ROI; Supplementary Fig. 8) on the central slice, alongside 3D analysis of pore volume and percentage porosity over 100 slices spanning approximately 1 mm in length around the central slice, using *Bruker CTAn v1.16.4.1 +* software (Sky-Scan 2003–2011, Bruker microCT 2012–2016). Outer, inner, and pore structure surface images were rendered in *Bruker CTAn* and visualised in *FEI Avizo Fire v8.1.1* (Konrad-Zeuse-Zentrum Berlin 1995–2014; FEI, SAS 1999–2014; Supplementary Note 7). One-way ANOVA and pairwise Student's t-tests with a Bonferroni correction were performed in PAST v3.1.1[55] to compare the pore density, average pore volume, and percent

porosity between eggshell morphotypes (Supplementary Table 11). Outliers (typically samples exhibiting imaging artifacts even after denoising) were excluded from these analyses.

## Ancient DNA extraction

Ancient DNA was extracted from 33 eggshell samples across each thickness from each location (north, south, and southwest; Fig. 1 and Supplementary Data 4). Samples were prioritised for DNA extraction based on their A/I ratio, with low A/I values preferred, as well as on their exposure to the environment at the time of collection, with those found buried prioritised. The same locality was not sampled twice for DNA in order to minimise the chance that two samples may have come from the same egg or same female. Ancient DNA was extracted from 200 mg of eggshell powder per sample in the Trace Advanced Ultra-Clean Environment (TrACE) at Curtin University, WA (Australia) between 2015 and 2018, following the protocol described by Dabney et al.[60] with minor changes (Supplementary Note 5), and in keeping with standard aDNA practice[61,62]. Shotgun sequencing libraries were prepared following the protocol described by Gansauge and Meyer[63] with minor changes (Supplementary Note 5 and Supplementary Table 1–3).

## Mitochondrial genome enrichment

3083 80-mer mitochondrial baits with 4X (20 bp) tiling were designed based on a consensus sequence of two published *Aepyornis*[1,2] (NCBI accession #KJ749824, and #AP014697, respectively) reference genomes, and one *Mullerornis* reference genome[1] (NCBI accession #KJ749825), and were manufactured through *MYcroarray*. Hybridisation enrichment of mitochondrial DNA was performed by following the MYbaits (*MYcroarray*) protocol (v.3, 2015) as per the manufacturer's instructions, with minor changes (Supplementary Note 5).

## DNA sequencing

Enriched libraries were quantified using a LabChip GX Touch HT (*Perkin Elmer*) following the manufacturer's instructions (Supplementary Note 5), and were pooled in equimolar concentrations in a total volume of 60 μl. To remove low-molecular weight primer dimer and library-build/capture artifacts, fragments between 140 bp and 300 bp were size-selected from the pooled library using two lanes of a Pippin Prep (*Sage Science*) eGel cassette following the manufacturer's instructions. The two lanes of size-selected library were recombined, and were purified and concentrated through a *QIAGEN* PCR Purification kit, following the manufacturer's instructions, with minor changes (Supplementary Note 5). The final sequencing library was quantified again on the LabChip GX Touch HT. The library was diluted to 4 nM in ultrapure water and was sequenced using *Illumina*'s high-throughput platform NextSeq, following the manufacturer's instructions with minor changes (Supplementary Note 5).

## Mitochondrial genome reconstruction

Sequences were trimmed using USEARCH v.8[64] and sequences below 30 bp in length were discarded, as they could not be meaningfully mapped to reference genomes. USEARCH v.8 was used to quality filter sequences (by employing an expected error rate of 1% of the length of the sequence), find unique sequences, and remove chimeric sequences (Supplementary Note 5 and Supplementary Data 4).

For each sample, sequences were iteratively mapped against a consensus elephant bird reference mitochondrial genome in Geneious v.8.1.6[65] using the default parameters under a 'medium-low sensitivity' option with 10 iterations. Mapped reads were then aligned to NCBI's GenBank reference database[66] using BLAST 2.2.30+[67] implemented through the Pawsey Centre's supercomputing facilities in order to obtain taxonomic assignments for the sequences. The blastn algorithm parameters evoked were as described by Grealy et al.[3]. Sequence taxonomy was assessed in MEGAN v.4.70.4[68] (Supplementary Note 5). To

remove potential contaminating sequences, reads aligning best to avian reference genomes were remapped onto the consensus genome generated from the last round of mapping, as before. A final strict consensus sequence with 50% majority-ruled based calling was generated, with positions having a coverage of <2 called as an 'N' and positions with no data represented by '?'. These final mitochondrial genomes can be found in GenBank (Accessions OP413790 - OP413810) or downloaded from DataDryad (https://doi.org/10.5061/dryad.3j9kd51nc). The authenticity of mapped reads was assessed by charting the frequency of nucleotide substitutions across reads in map-Damage 2.0.6[69,70] (Supplementary Fig. 1 and Supplementary Note 5).

## Phylogenetic inference

20 elephant bird mitochondrial genomes were aligned with two previously published elephant bird mitochondrial genomes[1,2] and eight outgroup ratites (Supplementary Table 4) using MAFFT v. 7.308[71] and MUSCLE v3.8.425[72] as implemented in Geneious v.8.1.6[65] using the default parameters (Supplementary Note 5). All protein-coding, rRNA and tRNA genes as well as the control region were extracted from the alignment and partitioned by codon position (protein-coding genes), and loops and stems (RNA genes). RCV and stemminess tests[73] were performed in PAUP v4a150[74] as described previously[3], with minor changes (Supplementary Note 5 and Supplementary Table 5–7). These tests were used to assess base composition bias and the extent of phylogenetic signal erosion in order to determine which partitions may benefit from RY coding that will alleviate the biases; however, none would benefit (Supplementary Note 5). The best-fitting substitution model for each partition was determined using jModelTest v.3.7[75,76] (Supplementary Note 5 and Supplementary Table 8). Mitochondrial phylogenetic trees were constructed on standard nucleotide-coded data, using maximum likelihood and Bayesian approaches implemented in RAxML v.1.5[77] and MrBayes v.3.2.6[78] (executed through the CIPRES v.3.3 online bioinformatic toolkit[79]), respectively (Supplementary Note 5). Tracer v1.6.1 was used to examine the convergence of Bayesian runs[80].

## Species delimitation

To determine whether the clades identified through phylogenetic analysis might represent different species, genetic distance within and between elephant bird specimens from each region exhibiting less than 10% missing data across 596 bp of cytochrome *oxidase I* (*COI*) were calculated in MEGA v.6.06[81] using the Kimura 2-parameter model[82] with pairwise deletion of missing data alongside default parameters for the remaining options (Supplementary Note 5). To gauge the limits of intra- and inter-specific variation in this barcoding region, the distance within and between genera of moa, rhea, emu, cassowary, and kiwi were also estimated in the same way using published sequences (Supplementary Data 5 and Supplementary Fig. 2). Species delimitation analysis[83] was also performed using the plugin (v.1.03) available within Geneious v. 10.0.5[65] (Supplementary Note 5 and Supplementary Table 9). Mantel tests comparing geographic distance and genetic distance matrices were conducted in R v1.3.1093[84].

## Molecular dating

Molecular dating was performed using MCMCTree[85] implemented in PAML v. 4.4d[86] as previously described[3] with minor changes (Supplementary Note 5 and Supplementary Data 6), including only the best representative elephant bird samples from each clade and incorporating previously published nuclear data[3]. Nine fossil-based age priors were used for calibration (Supplementary Note 5 and Supplementary Table 10).

## Palaeoproteomics

Protein extraction followed published protocols for ostrich eggshell proteomics analyses[87] at the Archaeobiomics laboratory at the

University of Turin (Italy), with protein digested using both trypsin and elastase (Supplementary Note 6). Eluted and dried down peptides were received at the Novo Nordisk Centre for Protein Research (Copenhagen, Denmark). Samples were separated on a 15 cm column (75 μm inner diameter) in-house laser pulled and packed with 1.9 μm C18 beads (Dr. Maisch, Germany) on an EASY-nLC 1200 (Thermo Fischer Scientific, Bremen, Germany) connected to a Q-Exactive HF-X (Thermo Fischer Scientific, Bremen, Germany) on a 77 min gradient. Resulting.raw files were searched using PEAKS v.8.5[88]. Parent ion and fragment ion mass tolerance were set to 10 ppm and 0.05 Da respectively, with unspecific digestion. Deamidation of N and Q, as well as oxidation of M, H, and W were set as variable PTMs. The files were searched against all available XCA-1 and XCA-2 protein sequences[21], and the common Repository of Adventitious Proteins (cRAP) to identify common contaminants. The proteomics datasets have been deposited to the ProteomeXchange Consortium via the Proteomics Identifications Database (PRIDE) partner repository with the dataset identifier PXD035725.

### Stable isotope analysis

Eggshell and vegetation samples (Supplementary Data 9) were prepared for isotopic analysis following the procedures outlined in Miller et al.[89] (Supplementary Note 8). $\delta^{13}C$, $\delta^{15}N$ and $\delta^{18}O$ were determined using an elemental analyser (NC 2500; CE Elantech, Lakewood, NJ) interfaced with Thermo Finnigan (San Jose, CA) Delta Plus XL or Delta V Plus mass spectrometers (Carnegie Institution of Washington, Washington, DC) (Supplementary Note 8). Isotopic data was also included from Hansford and Turvey[6]. Stable isotope values were corrected for enrichment from dietary sources and differences in atmospheric $CO_2$ (Supplementary Note 8). ISOERROR v1.04[90] was used to calculate the relative contribution of C3 and CAM vegetation to diet using source values of $\delta^{13}C$ from Crowley et al.[46] and 127 new plants collected from southwest Madagascar.

### Ancestral state reconstruction

Eggshell thickness, egg mass and bird mass from 65 birds[91] was used to produce phylogenetically-corrected regressions (Supplementary Note 9). Estimates of ancestral state for eggshell thickness, egg mass, and body mass were performed using *contMap*[92] within the *phytools* (v1.2-0) package in R v.4.2.0[84] using references for egg mass and eggshell thickness for palaeognath species (detailed in Supplementary Note 9 and Supplementary Code 1).

### Reporting summary

Further information on research design is available in the Nature Portfolio Reporting Summary linked to this article.

## Data availability

All supplementary methods and data related to this article can be found alongside the online version of the article, and on DataDryad (https://doi.org/10.5061/dryad.3j9kd51nc). Source data for Figs. 1, 2, 3 and 4 can be found in Supplementary Data 1–4, 6 and 10–11, 8 and 12, and 9. Mitochondrial genome sequences for the studied specimens can be found on GenBank (ncbi.nlm.nih.gov; accessions OP413790, OP413791, OP413792, OP413793, OP413794, OP413795, OP413796, OP413797, OP413798, OP413799, OP413800, OP413801, OP413802, OP413803, OP413804, OP413805, OP413806, OP413807, OP413808, OP413809, OP413810, KJ749824, KJ749825, AP014697, AP014698). Short read data has been deposited on NCBI's Short Read Archive (BioProject ID PRJNA880433). Proteomics data has been deposited on ProteomeXchange under the dataset identifier PXD035725. Eggshell specimens are currently housed at the University of Colorado (Boulder) and Curtin University (Western Australia), and will be donated to the University of Colorado Museum in 2023; in the meantime, requests for fossil material should be directed to G.M.

Correspondence and requests for other materials should be addressed to A.G.

## Code availability

Code used for ancestral state reconstruction can be found in Supplementary Code 1 and has also been deposited on DataDryad (https://doi.org/10.5061/dryad.3j9kd51nc).

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

## Acknowledgements

This work was funded by a grant from the Australian Research Council (ARC) awarded to JH (DE120100107). M.B. was supported in this research by an ARC future fellowship (FT0991741). G.F. wishes to acknowledge the National Geography Society, as well as the Easterbrook Distinguished Scientist Award (from the Division of Quaternary Geology and Geomorphology Division of the Geological Society of America), for funding sample collecting trips in Madagascar. B.D.'s work received support from the Ministry of University and Research (Young researchers "Rita Levi Montalcini"). KD's collections were made possible with funding from the National Science Foundation Graduate Research Fellowship Program, the P.E.O. Scholar Award, the Yale Institute of Biospheric Studies, the Yale MacMillan Center for International and Area Studies and the Yale Council on Archaeological Studies. Research permissions were granted by the Ministère de l'Enseignement Supérieur et de la Recherche Scientifique, Autorisation Numéro 128/13-MESupReS/SG/DGRP and by the Centre de Documentation et de Recherche sur l'Art et les Traditions Orales Malgaches (CEDRATOM), under the auspices of the Memorandum of Understanding between the University of Toliara, under the direction of Dr. Barthélémy Manjakahery, Director of the CEDRATOM, and Yale University, under the direction of Dr. Roderick McIntosh, Professor of Anthropology. Research permissions were also granted to Gifford Miller (University of Colorado, Boulder) by Director Jean-Aimé Rakotoarisoa of L'Institut de Civilisations – Musée d'Art et d'Archéologie de l'Université d'Antananarivo (2006–2007). Computing resources at the Pawsey Supercomputing Centre (WA) were used to perform BLAST searches. We would like to

thank Alison Devault at *MYcroarry* for assistance designing enrichment baits. We give special thanks to the Morombe Archaeological Project (MAP) team and to the people of Andavadoaka, Madagascar for helping collect eggshell. The authors acknowledge the facilities, and the scientific and technical assistance of the National Imaging Facility at the Centre for Microscopy, Characterisation and Analysis, The University of Western Australia—a facility funded by the University, State and Commonwealth Governments. M.M. and M.C. are supported by Danish National Research Foundation Award PROTEIOS (DNRF128). We thank Prof. Jesper Velgaard Olsen at the Novo Nordisk Center for Protein Research for providing access and resources, which were also funded in part by a donation from the Novo Nordisk Foundation (Grant No. NNF14CC0001). We would also like to acknowledge the passing of Professor Marilyn Fogel in 2022, and thank her for her contributions to stable isotope research.

## Author contributions

M.B., G.M. and J.Haile conceived the study. M.B. supervised the study. A.G. and M.B. designed the experiments. A.G. performed the genetic experiments. A.G. analysed the genetic data with assistance from M.P.; G.M., K.D. and J.Hansford collected and supplied eggshell for genetic analysis. G.M. performed the dating, amino acid racemisation and isotope experiments with S.C.; A.G. analysed the isotope data with assistance from G.M., S.C. and M.F.; K.D. performed dating. D.P. performed the micro-CT and 2D/3D analyses. A.G. analysed the micro-CT data. P.R. and A.H. performed confocal microscopy. B.D., M.C., M.M., J.Sakalauskaite and J.Stiller performed ancient protein experiments and analysed the data. L.J.L. performed body size calculations and ancestral state reconstructions with assistance from J.C.; A.G. wrote the manuscript with input from co-authors. Publication of this article was funded in part by the University of Colorado Boulder Libraries Open Access Fund.

## Competing interests

The authors declare no competing interests.

## Additional information

[1]Trace and Environmental DNA (TrEnD) Laboratory, School of Molecular and Life Sciences, Curtin University, Bentley, WA 6102, Australia. [2]The Australian National Herbarium, CSIRO, Bldg 502 Clunies Ross Street, Acton, ACT 2601, Australia. [3]INSTAAR and the Department of Geological Sciences, University of Colorado, Boulder, CO 80309, USA. [4]Vertebrate Evolution Group, School of Biology and Environmental Science, Queensland University of Technology, Brisbane QLD 4000, Australia. [5]Integrity Ag & Environment, 10511 New England Highway, Highfields QLD 4352, Australia. [6]EDGE Institute, University of California Riverside, 900 University Ave, Riverside, CA 92521, USA. [7]Centre for Microscopy, Characterisation and Analysis, Harry Perkins Institute of Medical Research, The University of Western Australia, Crawley, WA 6009, Australia. [8]National Imaging Facility, University of Western Australia, Crawley, WA 6009, Australia. [9]Department of Life Sciences and Systems Biology, University of Turin, Via Accademia Albertina, 13, 10123 Torino, Italy. [10]The Globe Institute, Faculty of Health and Medical Sciences, University of Copenhagen, Oster Farimagsgade 5, Bygning 7.101, 1353 Copenhagen, Denmark. [11]McDonald Institute for Archaeological Research, University of Cambridge, 2.4 West Tower, Downing St, Cambridge CB2 3ER, UK. [12]Novo Nordisk Foundation Center for Protein Research, Department of Biomedical Sciences, Faculty of Health and Medical Sciences, University of Copenhagen, Blegdamsvej 3B, 6.2, 2200 Copenhagen, Denmark. [13]Villum Centre for Biodiversity Genomics, Section for Ecology and Evolution, Department of Biology, University of Copenhagen, 2100 Copenhagen, Denmark. [14]Department of Geological Sciences, The University of Texas at Austin, 2275 Speedway Stop C9000, Austin, TX 78712, USA. [15]The Climate School, Columbia University, New York, NY 10025, USA. [16]Institute of Zoology, Zoological Society of London, Regent's Park, London NW1 4RY, UK. [17]Department of Biological Sciences, Northern Illinois University, DeKalb, IL, USA. [18]Department of Earth Sciences, University College London, Gower Street, London WC1E 6BT, UK. [19]Research Laboratory for Archaeology and the History of Art, Oxford University, Oxford OX12JD, UK. ✉e-mail: alicia.grealy@uqconnect.edu.au; gmiller@colorado.edu

