## [Peer Review File · Nature Communications]

Molecular exploration of fossil eggshell uncovers hidden lineage of giant extinct birdReviewers' Comments:

Reviewer #1:

Remarks to the Author:

In this manuscript, Grealy et al. characterize elephant bird egg shells and bone using a complex mixture of molecular, phylogenetic, stable isotope, and microCT analyses. This is an excellent paper and I look forward to it joining the literature. I have a few general comments, but will focus on some specific paleoproteomics questions.

Line 185/Figure 4b: It is difficult to distinguish the colors of the points with them all so light especially between the 10 and 15 values of nitrogen. Also, because of the difficulty in distinguishing the colors it is hard to see if the southern *Aepyornis* and *Mullerornis* specimens do have clear distinctions in carbon isotopes.

Line 371: Is the M to V substitution at position 58 observed in multiple samples from the southern and northern eggshells? Could this be an individual mutation from a single individual?

Figure 2: If the silhouettes in Figure 2 are derived from Phylopic, please acknowledge the artists who created them and their CC license.

Line 578-590: Mention that the protein extractions were digested with both trypsin and elastase in the main manuscript.

Supplemental SI 6.3 Data Analysis: Why wasn't a custom cleavage set designed in PEAKS for combined trypsin and elastase instead of using non-specific?

Figure S6: On a fast glance it is difficult to see the difference between the orange and green chains. If possible can the two species be split into individual ribbons?

Please make sure to put in the numbers for GenBank, ProteomeXchange, etc.

Reviewer #2:

Remarks to the Author:

Grealy and colleagues present a powerful multi-disciplinary study of the biology and diversity of Madagascar's extinct elephant birds. In particular, they use multiple lines of evidence to test some previously proposed taxonomic hypotheses (including the recent resurrection of the genus *Vorombe*), and refine our understanding of the diagnosis and distribution of different species. They provide compelling evidence for only three distinct elephant bird lineages — these data provide no evidence for the distinctness of *Vorombe titan* (the proposed largest bird species ever) at either the genus or species level. The authors also identify likely environmental drivers of elephant bird diversification, and reveal that gigantism in *Aepyornis* likely arose relatively rapidly. All very noteworthy and original advances in our understanding of these understudied and enigmatic birds.

This study is extremely rich in data, and the analyses have been executed thoroughly and documented comprehensively (the SI is especially detailed and well organised). The manuscript is clearly written (though there are some very long paragraphs that could benefit from being divided e.g. starting lines 95, 170, 239, 380), and the figures are clear and visually striking. Overall, this is a fascinating study that offers many important new insights into Madagascar's elephant birds — I enjoyed reading it very much and found it very thought-provoking.

While I have no issues with how any analyses were executed, I do have some concerns with how the interpretation of results is framed in one specific section (i.e. regarding the taxonomic identity of the

northern eggshells; see detailed comments below). To feel comfortable recommending this article for publication I would first need for the authors to either revise the framing of that section, or much more clearly present the evidence and arguments that underpin their current interpretation (in case I've misunderstood some important nuance). My other feedback is all relatively minor and should be easy to address. None of my recommendations require any further analyses.

With appropriate revisions, this excellent study would be highly appropriate for publication in Nature Communications. It's bound to be interesting to a wide audience. I would be happy to review a revised version of this manuscript.

Detailed feedback

Section: Evidence for hidden diversity within *Aepyornis* is recovered from eggshell in the far north #
The new data eggshell data from the north of Madagascar are very important for our understanding of elephant bird diversity and biology. However, I'm currently not convinced that there's much evidence to suggest the northern eggshell specimens belong to a taxon that's distinct from central Madagascan *Aepyornis hildebrandti*. This framing of this section of the manuscript currently implies that "novel taxon" is the null hypothesis, and the authors subsequently conclude that this null hypothesis cannot be rejected using their data. But I think the more appropriate null hypothesis (in the absence of any pre-existing data or taxonomic hypotheses about elephant birds from the north of Madagascar) is that the eggshells belong to one of the already recognised species — when framed that way, I don't think the authors' data allow them to reject synonymy with *A. hildebrandti*, which to me seems the more parsimonious explanation.

Unless I've misunderstood, the "medium" eggshells from the north appear anatomically indistinguishable from "medium" eggshells in the southwest (according to Figure 3a), and AD2667 from central Madagascar (i.e. *A. hildebrandti*) is specifically "of comparable thickness to the northern eggshell". Further, isolation by distance within a single taxon perfectly explains the reciprocal monophyly of the northern eggshell mtDNA with respect to the southwest "*hildebrandti*" lineage (plus population structure will lead to rejection of random coalescence within a single taxon, so IBD is not contradicted by the species delimitation results) — such a pattern is commonly observed in mtDNA from modern-day populations of "good species" with wide geographic ranges. The northern eggshells do exhibit a different isotopic composition from those in the centre, which implies the animals had a different diet, but that is not necessarily surprising given the specimens are from very different biomes — the animals from the north and centre could therefore just be exhibiting different realisations of a single fundamental niche. Many "good species" exhibit different isotopic signatures that reflect local resource availability and relative abundances.

Ultimately, I have no objection to the authors proposing that the northern eggshells might plausibly represent a distinct taxon as a hypothesis to be tested in the future (i.e. with skeletal data) — they certainly warrant further research in any case. But I strongly suggest the authors revise this section to remove the repeated strong implication that their taxonomic distinctness is a forgone conclusion (e.g. line 340 in particular implies that it must either be a distinct species or subspecies). I don't think the evidence as currently presented really supports that interpretation. In contrast, the alternative interpretation — i.e. that the geographic, dietary, and altitudinal range of *A. hildebrandti* was potentially much greater than previously thought — would be an important and interesting finding in its own right (that's fully consistent with the data).

Section: Recent divergence within *Aepyornis* is consistent with the rapid evolution of extreme gigantism

I find this part of the study very interesting and I agree with the authors' interpretation, but I found that it wasn't very clearly explained and attention is not drawn to several important pieces of evidence. Firstly, nowhere in this section is the reader explicitly directed to the relevant SI text and figures (from which we get the 54 kg estimate for the common ancestor of *Mullerornis* and *Aepyornis*).

And secondly, the most important point supporting the rapid and recent evolution of extreme gigantism is not highlighted — specifically, the inferred body mass for the common ancestor of *Aepyornis* ~1.4 million years ago is only 261 kg whereas Holocene individuals could be >1000 kg (Figure S10). I encourage the authors to add additional details here to support their argument.

SI 5.12 Molecular dating

The authors run their analyses with both an independent and autocorrelated rates model, but they consider only the results of the autocorrelated rates model because "this model biases against substantial rate shifts across short branches (e.g., deep among the palaeognath radiation)". However, an argument could be made that multiple substantial rate slowdowns deep in the tree are biologically realistic for palaeognaths, since it's been suggested that they convergently became large and flightless relatively early in their evolution (as noted later in SI 9). The results of the analyses using the independent rates model aren't currently viewable and I can't see any statement about exactly how they differ from those obtained using the autocorrelated rates model. Does the choice of model have much impact on the inferred ages of the different elephant bird lineages? If not, it would be good to simply add a brief statement to that effect to the SI text. If so, a stronger justification for preferring the results of autocorrelated rates model may be desirable.

Line 81

Reference 7 does not support the point being made on this line (the study is not about ratites). This is probably just a typo — reference 7 is cited more appropriately in the following sentence.

Line 106 - "...a mean thickness of more than 3.32 mm..." #
Delete "more than" — the mean thickness is exactly 3.32 mm.

Lines 146-141 - re: amino acid differences

I suggest the authors delete both instances of "substitution" from this sentence (but leave it otherwise unchanged). "Substitution" implies a directionality (i.e. that the residue does not represent the ancestral state) that is not evident from the comparisons made by the authors. Anyway, this does not affect the point being made.

Line 218 - "...cathermal..." #
This should be "cathemeral".

Line 231 (and similar uses elsewhere) - "...our data suggests..." #
"Data" is plural so it should more correctly be "...our data suggest..."

Line 325 - "Figure S3"
I think the authors are more correctly referring to Figure S2 in this sentence.

Line 421 - "...genetic diversity within Aepyornithidae is unprecedentedly low..." #
"Unprecedentedly" is perhaps not the most correct term here. Low genetic diversity is the precedent set by most palaeognath families, which generally include at most two species—Emeidae and Apterygidae are the only exceptions (with the latter being less analogous due to their much smaller body size and different ecology, though closer phylogenetic relationship). Specifically, I think the authors mean that diversity within Aepyornithidae is low compared to past taxonomic hypotheses based on skeletal anatomy.

Line 422-423 - "...a unique morphotype of eggshell found in Madagascar's far north..." #
According to what specific criterion are the northern eggshells morphologically unique from "medium" eggshells from further south? Figure 3a does not appear to support this conclusion, but there might be another line of evidence that I've overlooked.

Figure 1A

The figure caption mentions contour lines that show elevation, but I can't see these in my copy of the manuscript. Maybe a holdover from a previous version of the figure?

Figure 1B

The golden bars here help to make the top and bottom plots comparable, but it's unclear exactly what their thickness represents. The caption says they represent two standard deviations each side of the mean, but which mean exactly? If it's specifically the northern morphotype, why does the right hand bar extend out to >4.2mm when the thickest shell for that morphotype appears to be <3mm?

Figure 2B

Proapteryx micromeros is depicted as a closer relative to *Apteryx haasti* and *A. oweni* than to *A. australis*. In the molecular dating analysis it is used (correctly) to constrain the minimum age of the split between kiwi and elephant birds, so shouldn't *P. micromeros* more conservatively be depicted as the sister lineage to *Apteryx*?

Figure 3

The caption describes panels A, B, and C, but in my copy of the manuscript I can only see two panels (A and B).

Table S17

In footnote 1 there is still a placeholder reference to "Table SX" (I think it should be S17).

Data availability

In addition to submitting the mitochondrial consensus sequences to GenBank, I'd also encourage the authors to somehow make their raw sequencing data available.

Reviewer #3:

Remarks to the Author:

Comments on Molecular exploration of fossil eggshell uncovers hidden lineage of giant extinct bird by Grealy et al

This is a very thorough piece of work using multiple proxies to assess the diversity of aepyornithids in Madagascar. It uses eggshell morphology (thickness and porosity) with ancient DNA and proteins composition to assess diversity across a very large sample. It also relates the eggshell haplotypes to some skeletal haplotypes and thus the morphological skeletal diversity. Ultimately more data on skeletal diversity will be desirable but this is a very good analysis. It is certainly worth publishing but there are some minor points the authors should consider to improve the ms.

I refrain from commenting on the details of the ancient DNA and protein extraction methodology as am not an expert. I suggest though, that because the field of aDNA has moved so rapidly in recent years that the authors add the dates that the extractions of the DNA were done to the methods, as methods now would undoubtedly be different to those of a few years ago, and the date will in part explain why such and such was used or done.

Below, I mainly comment on the biology and taxonomy aspects of the paper.

Line 75 - is the reference 5 relevant to this point - seems ref 6 is more appropriate given it reviewed all species etc

Lines 80-81 - reference 7 is on ducks and so does not seem relevant to the point that homoplasy in post cranial skeletons poorly distinguishes ratite taxa - suspect authors mean refs 8 and 9, suggest reorder latter as 7 and 8.

Line 84 - suggest reorder refs so duck paper currently 7, becomes 9, then it will apply here

Line 87 - regarding skeletal fossils are scarce, I do not think this is correct; many thousands have been collected as recorded by the many works of Monnier, Lambertson and earlier authors; but most of those in Madagascar were destroyed when the museum burnt in the 1990s; nevertheless there are many bones in France (Muséum national d'Histoire naturelle; in Lyon and other places), Austria in Vienna, Oslo and scattered in numerous other museums across the world, as partly listed and used by Hansford and Turvey. Most such bones were from swamps and not the dunes where the eggshell is recovered from.

Lines 98-99 - with all previously dated specimens is incorrect. Only with some dated specimens - there are sites with aepyornithid bones that are dated up to 20kyr, for example your ref 38 lists multiple sites in the late Pleistocene with aepyornithids which are thus much older than the eggshell dates. What is meant by coincide? I am not aware that bones are found with the eggshell. Yes, there are swamp deposits that have bones in them in the same southern region for example but if there are sites with both bones and shell in them then these should be detailed or at least the references listed. But I think what is meant that in the same general region there are dune deposits and swamp deposits of similar age with elephant bird remains.

Line 100 - regarding 'morphologically' - think what is meant is 'from skeletal material sourced from the same geographic areas'

Line 102 - regarding becoming extinct soon after suggest rewrite as 'elephant birds extinction dates more recently' [says what you mean and lacks the imprecision of the word soon which might mean 2 years or 500 years as even the latter is very short in a geological context]

Lines 104-106 - I note that the ref by Gill revealed that dune sites usually revealed a bimodal distribution for moa eggshell, yet 4 species were represented. [Gill, B. J. Thickness histograms of Holocene fossil eggshell fragments indicate diversity and relative abundance of moas (Aves: Dinornithiformes) at North Island sites. New Zealand Journal of Zoology 49, 143-165 (2021).]

Line 107 - regarding the phrase 'represents hitherto undescribed fossils' - think what is meant is 'represent fossils from a new locality'; each eggshell fragment is a fossil and it is premature at this stage in the ms to conclude they relate to a unique molecular clade of aepyornithid; or is it that they represent a previously undescribed type of fossil eggshell?

Lines 112-115 - I would prefer ranges with means and confidence interval as more biologically relevant. Also, what is relevant for the construction of the algorithm is the mass of the incubating bird, thus the large size of male fowl is irrelevant and likewise the large size of female ratites is irrelevant as it is the smaller males that do the incubation - hence it is their mass that should be used. Please confirm appropriate masses for taxa were used.

Line 121-122 - Re 'We find that eggshell and skeletal specimens cluster into' I suggest edit this to say "We find that DNA haplotypes of eggshell and skeletal specimens cluster.."

Line 134 - re 'confirm the description of Mullerornis as the more gracile genus' - No - what it confirms is that the thin eggshell less than 1.5 mm is associated with the well-known genus of more gracile birds, Mullerornis. [after all the genus is based on a species Mullerornis betsilei based on bones and other taxa M. agilis, M. rudis and M. grandis and Aepyornis modestus - all based on bones-are attributed to this genus now synonymised to form one monotypic genus with the species M. modestus (Milne-Edwards & Grandidier, 1869)]

Line 141 - re families here - But for example the Howard and Moore Checklist does not have these at family level, so say who does, because it must be a quite recent change even by Clements...

what is the equivalent genetic distance between the three moa families (Megalapterygidae, Dinornithidae and Emeidae)? that might be a better comparison given they have some diversity.

Line 142 – re 'into a new monogeneric family..' - the family was established by Lamberton in 1934, so it is not new, rather you find support for the original hypothesis.

Line 155 – Figure S4 - I think you mean Figure S3, there is nothing indicated in S4 or S5 to support this stmt.

Line 183 – Table S9. you mean Table S19; however, this table does not give summary data for carbon nitrogen and oxygen and such would be useful rather than just the results of pairwise comparisons. Perhaps put key exemplary mean and range values here in the text.

Line 186 – says significantly more negative – there may be statistical significance but there is a very wide overlap as for all the data the means are -26.8 (-31.2 to -23.9) vs -25.96 (-29.4 to -21.4).

Line 221 – why hav syn. *M. agilis* in brackets - the synonymy under *M. modestus* includes *M. betsilei*, *M. rudis* and *M. grandis* as well - so why mention just 1 nomen.

Line 235-236 – eggshell from central Madagascar remains to be tested - both eggshell and bones remain to be DNA identified.

Lines 236-237 – the sentence beginning 'Prior description of numerous smaller mullerornithid..' – There are only 5 Mullerornis taxa named. The suggestion re sexual dimorphism and growth series is possible, but more likely, is that as these deposits are not all late Holocene and some are up to mid 20ka, that there will be temporal variation in size (related to mean temp) and this very likely explains the mean larger size of the population named as grandis. Note various of the moa taxa [eg *Pachyornis geranoides* (=mappini), *Euryapteryx curtus*] varied greatly (~50% reduction in size from late glacial to late Holocene; and over latitude in the Late Holocene alone; *Dinornis* populations comprise birds less than half the mean size only 200 km from the Canterbury Plains during the same late Holocene period, but at >600m altitude in montane zones) for example, see data in:

Worthy, T.H., 1987. Sexual dimorphism and temporal variation in the North Island moa species *Euryapteryx curtus* (Owen) and *Pachyornis mappini* Archey. *National Museum of New Zealand Records* 3(6): 59-70.

Worthy, T.H., M. Bunce, A. Cooper, P. Scofield, 2005. *Dinornis* – an insular oddity, a taxonomic conundrum reviewed. Alcover, J. A. & Bover, P. (eds), *Proceedings of the International Symposium "Insular Vertebrate Evolution: The Palaeontological approach"* Monographies de la Societat d'Historia Natural de les Balears 12: 337-390. and in *Lost World of the Moa*.

Lines 248-249 – This has a parallel in the largest individuals of the large *Dinornis robustus*, where a few big members of the larger sex stand out, see the 2005 paper just mentioned (and some of these were given taxon names, eg *maximus*, *excelsius*, and for the North Island species e.g., *hercules*) because folk have a tendency to give unwarranted value to the extremes of +20% size range around the mean in a large animal than in a small one. So, more consideration of whether these larger animals named Vorombe were just large females was definitely warranted before they were named.

Line 277-278 – re the stmt 'advocate synonymising Vorombe titan and *Aepyornis maximus*..' – to be precise, you mean "advocate returning the taxon *Aepyornis titan* Andrews 1894 back into synonymy with *A. maximus* Geoffroy Saint-Hilaire, 1851; and in doing so we synonymise Vorombe Hansford and Turvey 2020 with *Aepyornis* Geoffroy Saint-Hilaire 1851". NOTE Vorombe only became available in 2020 (the correction), NOT 2018.

The reference is: Hansford J.P., S. T. Turvey, 2020. Correction to 'Unexpected diversity within the extinct elephant birds (Aves: Aepyornithidae) and a new identity for the world's largest bird'.

<https://doi.org/10.1098/rsos.201358>

Lines 282-283 - I wholly agree with the suggestion that titan may be the females – but it is worth drawing the analogy with *Dinornis robustus* where data show a large tail of huge forms in the large sample size available for Canterbury birds (see the 2005 paper mentioned above).

Lines 284-285 – rewording is required. This is now dated; first, the refs 8 and 9 separated the North and the South Island populations into two species - so these taxa listed by the authors here belong *sensu stricto* only to the North Island *Dinornis* and so you should add North Island to this sentence; and it is *novaezealandiae*; so for North Island *Dinornis* these taxa became *D. novaezealandiae*, and in the South Island bones called these same 3 taxa became *D. robustus*. But, for example, in the 3-*Dinornis* system of nomenclature (in vogue 1980s-1990s) multiple taxa were already in synonymy so in the South Island, some 7 taxa including *D. torosus*, *D. maximus*, are synonyms of the single South Island *Dinornis robustus* – see the NZ checklist for correct synonymies.

Line 288 – re sexual dimorphism – suggest cite the papers on moas given above as they are the most pertinent and closest in terms of variation shown with data eg Worthy et al 2005, Worthy 1987, Worthy and Holdaway 2001.

Lines 297-299 – there is a paper on histology of aepyornithid bones..but no medullary bone was identified in the sample.

Lines 310-311 – comments re diversity - Madagascar is only 1 island - moa diversity was also driven by allopatry via islands to a large degree and only 3 to 4 species ever lived together - so 4 taxa in Madagascar vs 6 in NZs SI is not greatly different and the lower diversity in Madagascar is likely explained to a great degree by other browsers being present (all the lemurs for example).

Line 313 – the pre-Holocene skeletal record for elephant birds is scant – suggest refer to see Burney et al 2004 (A chronology for late prehistoric Madagascar) for preholocene dates, eg. Ampasamazimba (22 ka), Andrakibaikely Pond, Ankazomiranga (12-30 ka), Antsirabe (17-20 ka), and for Tsaramody (see Samonds et al 2019) =- so not really scant - were hundreds of ebird bones from Antsiribe...

Lines 331-332 - Is it possible that some of the distinction (between the Northern and central) derives from temporal discordance as the central ones could be late glacial age, apart from the 1000 km allopatry - which has led to far great disparity in contemporary kiwi and moa populations?

Line 340 – re a new species – you mean an undescribed species.

Line 342-343 – re *hildebrandti* - need a reference for this - I suspect you are citing Hansford and Turvey - yet they also listed *Belo-sur-Mer* which is on the coast at sea level. But you need to add also Samonds et al 2019 here.

Line 410 – I do not agree with use of the word paucity - actually have been hundreds of specimens, still are hundreds in France, I rather think a 19th century philosophy of small metric differences being used to characterise taxa (as for moa) and that there have been no revisions since the 1930s except for 1 study underpin the issues. ie actual biological variation expected of a species was not applied to the taxonomy until very recently.

Line 423-424 – represents a novel lineage – need to add one very close to *hildebrandti*

Line 425 – we propose a revision – this does not encapsulate that above you do synonymise *A. titan* with *A. maximus* once more and thus *Vorombe* with *Aepyornis*.

Figure 2 caption - The numbers 15 and 16 as superscripts on these do not match anything obvious and are not explained in the caption.

Figure 4 caption - Opposite the panel for A are 4 sample types, yet only 3 are in panel A - is Central missing in error? The point clouds in B are not overly clear. The 5 circled ranges on the C3 half of the figure are not well enough explained and not very relatable to the habitat types in the key

Line 605 – two authors have initial JH

References - the references in SI 10 are in numeral form so cannot be related to the references listed in SI 11.

Table 1 is confusing as the references are not well associated with the taxa. A species is described only once, though there may be extra morphological descriptions later. Strongly suggest to add the authors and date to the taxon names so they are complete taxon names; so eg *A. hildebrandti* Burckhardt 1893; Note - Vorombe titan was NOT described by Hansford and Turvey 2018 - the species is Vorombe titan (Andrews 1894) and Hansford and Turvey in 2020 (not 2018) (<https://doi.org/10.1098/rsos.201358>) erected the genus name Vorombe for *A. titan* Andrews 1894.

Trevor Worthy - PS if you need help with the references please ask.

We thank the reviewers for their constructive feedback. We have now endeavoured to take on board or address each comment below. Line numbers refer to the revised manuscript and changes have been highlighted by red text in the manuscript. We believe the reviewer's comments have helped us improve the manuscript significantly and hope the revised submission is now acceptable for publication in *Nature Communications*.

REVIEWER COMMENTS

Reviewer #1

(1) In this manuscript, Grealy et al. characterize elephant bird egg shells and bone using a complex mixture of molecular, phylogenetic, stable isotope, and microCT analyses. This is an excellent paper and I look forward to it joining the literature. I have a few general comments, but will focus on some specific paleoproteomics questions.

We thank the reviewer for their constructive feedback and have endeavoured to take on or address their points below.

*(2) Line 185/Figure 4b: It is difficult to distinguish the colors of the points with them all so light especially between the 10 and 15 values of nitrogen. Also, because of the difficulty in distinguishing the colors it is hard to see if the southern *Aepyornis* and *Mullerornis* specimens do have clear distinctions in carbon isotopes.*

We have now increased the opacity—and decreased the diameter—of the dot points in Figure 4b so that it is easier to distinguish between specimen data points, as suggested by the reviewer. We have also boldened the centroid values for each taxon larger for clarity, and added asterisks to highlight the significant differences between stable isotope values.

(3) Line 371: Is the M to V substitution at position 58 observed in multiple samples from the southern and northern eggshells? Could this be an individual mutation from a single individual?

The M → V substitution is present in two out of three samples from the southern thick morphotype (PALTO 86 and PALTO 96), the third sample (PALTO 85) does not have this position covered. The sequences in the alignment represent the consensus of the three. We have now modified Figures S4 and S5 in the Supplementary Information to make it clearer that any mutations observed were present in at least two out of three individuals.

(4) Figure 2: If the silhouettes in Figure 2 are derived from Phylopic, please acknowledge the artists who created them and their CC license.

The silhouettes in Figure 2 were constructed by AG using Adobe Illustrator and do not require a CC license.

(5) Line 578-590: Mention that the protein extractions were digested with both trypsin and elastase in the main manuscript.

We have now mentioned that protein was “digested using both trypsin and elastase (SI 6.1)” at Lines 604-605 in the main manuscript, as suggested by the reviewer.

(6) Supplemental SI 6.3 Data Analysis: Why wasn't a custom cleavage set designed in PEAKS for combined trypsin and elastase instead of using non-specific?

In ancient samples, hydrolytic damage due to natural diagenesis is expected, as much as cleavages due to the action of the enzymes. Therefore, a non-specific search is used to maximise recovery. We have now added this explanation to SI 6.3 to answer the reviewer's question.

(7) Figure S6: On a fast glance it is difficult to see the difference between the orange and green chains. If possible can the two species be split into individual ribbons?

We appreciate the reviewer's suggestion; however, this was a deliberate choice, made to highlight those regions where substitution does result in predicted structural variation. The failure to discriminate between them illustrates that they show no difference in structural prediction in these regions of the proteins.

(8) Please make sure to put in the numbers for GenBank, ProteomeXchange, etc.

Mitochondrial DNA sequences have now been uploaded to GenBank and the accessions are now listed at Lines 572 and 680 (Accession numbers: OP413790-OP413810). Fastq files are available on NCBI's Short Read Archive (BioProject ID PRJNA880433, Submission ID 12049986)—see Line 681. Other data has been deposited on DataDryad (<https://doi.org/10.5061/dryad.3j9kd51nc>)—see Line 682. Proteomics data has been deposited on ProteomeXchange under the number PXD035725, which is now provided at Lines 616 and 683 of the main manuscript.

Reviewer #2

(9) *Grealy and colleagues present a powerful multi-disciplinary study of the biology and diversity of Madagascar's extinct elephant birds. In particular, they use multiple lines of evidence to test some previously proposed taxonomic hypotheses (including the recent resurrection of the genus *Vorombe*), and refine our understanding of the diagnosis and distribution of different species. They provide compelling evidence for only three distinct elephant bird lineages — these data provide no evidence for the distinctness of *Vorombe titan* (the proposed largest bird species ever) at either the genus or species level. The authors also identify likely environmental drivers of elephant bird diversification, and reveal that gigantism in *Aepyornis* likely arose relatively rapidly. All very noteworthy and original advances in our understanding of these understudied and enigmatic birds.*

This study is extremely rich in data, and the analyses have been executed thoroughly and documented comprehensively (the SI is especially detailed and well organised). The manuscript is clearly written (though there are some very long paragraphs that could benefit from being divided e.g. starting lines 95, 170, 239, 380), and the figures are clear and visually striking. Overall, this is a fascinating study that offers many important new insights into Madagascar's elephant birds — I enjoyed reading it very much and found it very thought-provoking.

Following the reviewer's suggestion, we have divided the first paragraph of the Results and Discussion (starting Line 95) into two paragraphs, the second beginning at Line 118. The paragraph that began at Line 170 has now been split at Line 183. The paragraph that began at Line 239 has now been split at Line 256. The paragraph that began at Line 380 has now been split at Line 299.

(10) *While I have no issues with how any analyses were executed, I do have some concerns with how the interpretation of results is framed in one specific section (i.e. regarding the taxonomic identity of the northern eggshells; see detailed comments below). To feel comfortable recommending this article for publication I would first need for the authors to either revise the framing of that section, or much more clearly present the evidence and arguments that underpin their current interpretation (in case I've misunderstood some important nuance). My other feedback is all relatively minor and should be easy to address. None of my recommendations require any further analyses.*

With appropriate revisions, this excellent study would be highly appropriate for publication in Nature Communications. It's bound to be interesting to a wide audience. I would be happy to review a revised version of this manuscript.

We are thrilled the reviewer enjoyed reading the manuscript and thank them for their constructive feedback. We have addressed the reviewer's concerns regarding the interpretation of some results (see Comment 13) and provide a response below.

(11) # Section: *Evidence for hidden diversity within *Aepyornis* is recovered from eggshell in the far north* # *The new data eggshell data from the north of Madagascar are very important for our understanding of elephant bird diversity and biology. However, I'm currently not convinced that there's much evidence to suggest the northern eggshell specimens belong to a taxon that's distinct from central Madagascan *Aepyornis hildebrandti*. This framing of this section of the manuscript currently implies that "novel taxon" is the null hypothesis, and the authors subsequently conclude that this null hypothesis cannot be rejected using their data. But I think the more appropriate null hypothesis (in the absence of any pre-existing data or taxonomic hypotheses about elephant birds from the north of Madagascar) is that the eggshells belong to one of the already recognised species — when framed that way, I don't think the authors' data allow them to reject synonymy with *A. hildebrandti*, which to me seems the more parsimonious explanation.*

Unless I've misunderstood, the "medium" eggshells from the north appear anatomically

indistinguishable from "medium" eggshells in the southwest (according to Figure 3a), and AD2667 from central Madagascar (i.e. *A. hildebrandti*) is specifically "of comparable thickness to the northern eggshell". Further, isolation by distance within a single taxon perfectly explains the reciprocal monophyly of the northern eggshell mtDNA with respect to the southwest "hildebrandti" lineage (plus population structure will lead to rejection of random coalescence within a single taxon, so IBD is not contradicted by the species delimitation results) — such a pattern is commonly observed in mtDNA from modern-day populations of "good species" with wide geographic ranges. The northern eggshells do exhibit a different isotopic composition from those in the centre, which implies the animals had a different diet, but that is not necessarily surprising given the specimens are from very different biomes — the animals from the north and centre could therefore just be exhibiting different realisations of a single fundamental niche. Many "good species" exhibit different isotopic signatures that reflect local resource availability and relative abundances.

Ultimately, I have no objection to the authors proposing that the northern eggshells might plausibly represent a distinct taxon as a hypothesis to be tested in the future (i.e. with skeletal data) — they certainly warrant further research in any case. But I strongly suggest the authors revise this section to remove the repeated strong implication that their taxonomic distinctness is a forgone conclusion (e.g. line 340 in particular implies that it must either be a distinct species or subspecies). I don't think the evidence as currently presented really supports that interpretation. In contrast, the alternative interpretation — i.e. that the geographic, dietary, and altitudinal range of *A. hildebrandti* was potentially much greater than previously thought — would be an important and interesting finding in its own right (that's fully consistent with the data).

We agree with the reviewer that we cannot reject synonymy with *A. hildebrandti*. This is why we have used the phrases "unique lineage" and "hidden diversity" or "evolutionary significant unit", and why we suggest that "IBD cannot be ruled out" (Line 346), that "gene flow [may have been] maintained by male dispersal" (Line 348-349), and that "whether this taxon can be considered a novel species...cannot be concluded based on these data" (Line 352-353). As suggested by the reviewer, we have now rephrased this section to reinforce the uncertainty in the identification of this eggshell:

Line 339: "further supporting the idea that" has been changed to "which may suggest"

Line 352-353: "can be considered a novel species or merely a subspecies" has been changed to "can be considered a novel species, subspecies, or merely a population of *A. hildebrandti*"

Line 361 and Line 364-365: "northern taxon" has been changed to "northern elephant birds" and "members of the northern clade"

Line 368-370: We have added the phrase, "However, it may instead suggest that *A. hildebrandti* was a generalist species with the ability to subsist in a variety of environments"

Line 445: We have now added "(likely *A. hildebrandti*)" to the conclusion after "represents a novel lineage of *Aepyornis*".

(12) # Section: Recent divergence within Aepyornis is consistent with the rapid evolution of extreme gigantism #I find this part of the study very interesting and I agree with the authors' interpretation, but I found that it wasn't very clearly explained and attention is not drawn to several important pieces of evidence. Firstly, nowhere in this section is the reader explicitly directed to the relevant SI text and figures (from which we get the 54 kg estimate for the common ancestor of Mullerornis and Aepyornis). And secondly, the most important point supporting the rapid and recent evolution of extreme gigantism is not highlighted — specifically, the inferred body mass for the common ancestor of Aepyornis ~1.4 million years ago is only 261 kg whereas Holocene individuals could be >1000 kg (Figure S10). I encourage the authors to add additional details here to support their argument.

As per the reviewer's suggestion we have now explicitly directed the reader to the relevant SI text and figures at Line 416 and 417-418. We have also now added further details to explain at Lines 414-419: "The extremely large body size (ca. 700-1000 kg) in *Aepyornis maximus* appears to be a derived trait, with the crown elephant birds estimated to be closer to the size of *Mullerornis* (ca.

80 kg; SI 9, Figure S12), and the common ancestor of *Aepyornis* being half that size (ca. 400-500 kg) only a mere 1.4 my ago (SI 9, Figure S12). That is, body size of birds in the lineage leading to *Aepyornis maximus* nearly doubled between the Mid- to Late Pleistocene.”

(13) # SI 5.12 Molecular dating # The authors run their analyses with both an independent and autocorrelated rates model, but they consider only the results of the autocorrelated rates model because "this model biases against substantial rate shifts across short branches (e.g., deep among the palaeognath radiation)". However, an argument could be made that multiple substantial rate slowdowns deep in the tree are biologically realistic for palaeognaths, since it's been suggested that they convergently became large and flightless relatively early in their evolution (as noted later in SI 9). The results of the analyses using the independent rates model aren't currently viewable and I can't see any statement about exactly how they differ from those obtained using the autocorrelated rates model. Does the choice of model have much impact on the inferred ages of the different elephant bird lineages? If not, it would be good to simply add a brief statement to that effect to the SI text. If so, a stronger justification for preferring the results of autocorrelated rates model may be desirable.

Analyses with both independent and autocorrelated rates models have been deposited on DataDryad (<https://doi.org/10.5061/dryad.3j9kd51nc>); we have also added a summary table to the Supplementary information (Table S16) that summaries the differences in the results, and we briefly discuss them. While there are differences, we still preference the results of the autocorrelated model, and as per the reviewer's suggestion, have now added additional justification to the Supplementary Information (SI 5.12), which is as follows:

“Although there are likely to be multiple independent rate decelerations among ratites, the *a priori* expectation would be that these would not occur along the short basal-most lineages, but the longer lineages leading to large and/or long-lived ostrich, rheas, moa, elephant birds, etc. (since rate correlates like size and longevity are predicted to shift along these lineages). Further, ratograms suggest that there is a somewhat bimodal distribution of rates across the branches of the tree: slow rates among the flightless bird tips and their shallow stems, then faster rates among outgroups and among deeper palaeognaths. The independent (lognormal) rates distribution is a poor fit for a distribution that is somewhat bimodal, whereas such as distribution does not violate the autocorrelated rates model. For this reason, dates estimated using the autocorrelated rates model was presented in Figure 2, but all analyses can be found on DataDryad (see “Data availability statement” in the main text). The results of both are also summarised in Table S16 below.”

(14) # Line 81 # Reference 7 does not support the point being made on this line (the study is not about ratites). This is probably just a typo — reference 7 is cited more appropriately in the following sentence.

We have now exchanged Reference 7 and Reference 8, such that Reference 8 is now Bunce et al. 2003 and Reference 9 is now Mitchell et al. 2014. The in-text referencing has been updated at Lines 81, 84, 289, 292, and 294.

(15) # Line 106 - "...a mean thickness of more than 3.32 mm..." # Delete "more than" — the mean thickness is exactly 3.32 mm.

The words “more than” has now been deleted from Line 107 as suggested by the reviewer.

(16) # Lines 146-141 - re: amino acid differences # I suggest the authors delete both instances of "substitution" from this sentence (but leave it otherwise unchanged). "Substitution" implies a directionality (i.e. that the residue does not represent the ancestral state) that is not evident from the comparisons made by the authors. Anyway, this does not affect the point being made.

Both instances of “substitution” have now been deleted from the paragraph between Lines 147-163, as suggested by the reviewer; they are now referred to as “differences”.

(17) # Line 218 - "...cathermal..." # This should be "cathemeral".

The spelling error “cathermal” has now been corrected to “cathemeral” at Line 222, as suggested by the reviewer.

(18) # Line 231 (and similar uses elsewhere) - "...our data suggests..." # "Data" is plural so it should more correctly be "...our data suggest..."

We have now corrected the instance of “data suggests” to “data suggest” at Line 235, as suggested by the reviewer. We have also changed “supports” to “support” at Line 237, and “shows” to “show” at Line 247.

(19) # Line 325 - "Figure S3" I think the authors are more correctly referring to Figure S2 in this sentence.

The reviewer is correct; we have now changed the reference from “Figure S3” to “Figure S2” at Line 337. Other instances of this error have also been updated throughout the text.

(20) # Line 421 - "...genetic diversity within Aepyornithidae is unprecedentedly low..." # "Unprecedentedly" is perhaps not the most correct term here. Low genetic diversity is the precedent set by most palaeognath families, which generally include at most two species—Emeidae and Apterygidae are the only exceptions (with the latter being less analogous due to their much smaller body size and different ecology, though closer phylogenetic relationship). Specifically, I think the authors mean that diversity within Aepyornithidae is low compared to past taxonomic hypotheses based on skeletal anatomy.

Here, we are referring to the fact that mitochondrial divergence within Aepyornithidae is low, even compared to other large ratites, with the genetic distance between the two closest species being lower than any other ratite (see Discussion Lines 389-390). However, the reviewer also makes a good point that the taxonomic diversity is low compared with past estimates; we have now rephrased this section (Lines 440-442) as per the reviewer’s suggestion to include both perspectives: “diversity within Aepyornithidae is low (with regard to mitochondrial divergence relative to other large ratites, and compared with past taxonomic hypotheses based on skeletal morphology), with skeletal morphs potentially instead representing extreme sexual dimorphism”.

(21) # Line 422-423 - "...a unique morphotype of eggshell found in Madagascar’s far north..." # According to what specific criterion are the northern eggshells morphologically unique from "medium" eggshells from further south? Figure 3a does not appear to support this conclusion, but there might be another line of evidence that I've overlooked.

The reviewer is correct, while the northern eggshell would be considered unique from the morphotypes in the far south (based on the average thickness being significantly different), there is no evidence that it is a unique morphotype from the central eggshell (though we did not have a large collection of central eggshell for comparison). We have now rephrased this statement (Line 443) as “a new eggshell collection found in Madagascar’s far north...” for clarity.

(22) # Figure 1A # The figure caption mentions contour lines that show elevation, but I can't see these in my copy of the manuscript. Maybe a holdover from a previous version of the figure?

The reviewer is correct; contour lines were deleted from a previous version of the figure because the figure appeared very crowded. We have now deleted this statement from the caption of Figure 1a; however, we can also replace the contour lines if desired by the editorial team or reviewers.

(23) # Figure 1B # The golden bars here help to make the top and bottom plots comparable, but it's

unclear exactly what their thickness represents. The caption says they represent two standard deviations each side of the mean, but which mean exactly? If it's specifically the northern morphotype, why does the right hand bar extend out to >4.2mm when the thickest shell for that morphotype appears to be <3mm?

The bars represent 95% confidence intervals of the mean thicknesses for each morphotype. The right-hand bar for the northern morphotype does only extend to about 2.8 mm, it just overlaps with the left-hand bar for the southern morphotype. The mean is represented by the silhouette which is positioned over the X-axis at the mean thickness, and the width of the silhouette is also scaled to thickness. However, we have now altered this figure (Figure 1b) for clarity, as suggested by the reviewer: silhouettes and bars have been moved directly above their respective distributions. In addition, the bars have been recoloured to be consistent with the eggshell morphotypes plotted on the map, and with the stable isotope figure (Figure 4).

(24) # Figure 2B # Proapteryx micromeros is depicted as a closer relative to Apteryx haasti and A. oweni than to A. australis. In the molecular dating analysis it is used (correctly) to constrain the minimum age of the split between kiwi and elephant birds, so shouldn't P. micromeros more conservatively be depicted as the sister lineage to Apteryx?

The reviewer is correct; we have now altered the figure (Figure 2b) to place *P. micromeros* as sister to the *Apteryx* lineage, as suggested.

(25) # Figure 3 # The caption describes panels A, B, and C, but in my copy of the manuscript I can only see two panels (A and B).

The reviewer is correct; the cross-sectional diagram of the eggshell structure was present in an earlier version of the figure but was deleted to simplify the figure. However, this can be re-introduced if desired by the editorial team or reviewers. Mention of this panel has now been deleted from the Figure 3 caption.

(26) # Table S17 # In footnote 1 there is still a placeholder reference to "Table SX" (I think it should be S17).

We have now replaced the placeholder reference of “Table SX” in the footnote with “Table S19” (the most updated label for this table), as suggested by the reviewer.

(27) # Data availability # In addition to submitting the mitochondrial consensus sequences to GenBank, I'd also encourage the authors to somehow make their raw sequencing data available.

Raw sequencing data has now been uploaded to NCBI's short read archive and can be accessed under the accession: (BioProject ID PRJNA880433, Submission ID 12049986). This is now mentioned at Line 681 in the Data Accessibility Statement.

Mitochondrial DNA sequences have now been uploaded to GenBank and the accessions are now listed at Lines 572 and 680 (Accession numbers: OP413790-OP413810). Other data has been deposited on DataDryad (<https://doi.org/10.5061/dryad.3j9kd51nc>), which is mentioned at Line 682.

Reviewer #3

(28) This is a very thorough piece of work using multiple proxies to assess the diversity of aepyornithids in Madagascar. It uses eggshell morphology (thickness and porosity) with ancient DNA and proteins composition to assess diversity across a very large sample. It also relates the eggshell haplotypes to some skeletal haplotypes and thus the morphological skeletal diversity. Ultimately more data on skeletal diversity will be desirable but this is a very good analysis. It is certainly worth publishing but there are some minor points the authors should consider to improve the ms.

We thank the reviewer for their constructive feedback and have endeavoured to take on or address their points below.

(31) I refrain from commenting on the details of the ancient DNA and protein extraction methodology as am not an expert. I suggest though, that because the field of aDNA has moved so rapidly in recent years that the authors add the dates that the extractions of the DNA were done to the methods, as methods now would undoubtedly be different to those of a few years ago, and the date will in part explain why such and such was used or done. Below, I mainly comment on the biology and taxonomy aspects of the paper.

We have now mentioned that aDNA was extracted from eggshell between 2015 and 2018 at Line 538, as suggested by the reviewer.

(32) Line 75 - is the reference 5 relevant to this point - seems ref 6 is more appropriate given it reviewed all species etc

This reference was cited because it specifically made the point that “few of the subsequent descriptions [of elephant birds] include information on cranial material” and note that “the skull of *Aepyornis* was not discovered until approximately 40 years following the initial description of the taxon” (Balanoff and Rowe 2007). However, we now include Reference 6 and 7 in addition to Reference 5 at Line 75, as per the reviewer’s suggestion.

(33) Lines 80-81 - reference 7 is on ducks and so does not seem relevant to the point that homoplasmy in post cranial skeletons poorly distinguishes ratite taxa – suspect authors mean refs 8 and 9, suggest reorder latter as 7 and 8.

The reviewer is correct and this is also mentioned by Reviewer 2 (Comment 14); we have now exchanged Reference 7 and Reference 8, such that Reference 8 is now Bunce et al. 2003 and Reference 9 is now Mitchell et al. 2014. The in-text referencing has been updated throughout.

(34) Line 84 - suggest reorder refs so duck paper currently 7, becomes 9, then it will apply here

Following from Comment 14 and 33 above, Mitchell et al. 2014 is now Reference 9.

(35) Line 87 – regarding skeletal fossils are scarce, I do not think this is correct; many thousands have been collected as recorded by the many works of Monnier, Lambertson and earlier authors; but most of those in Madagascar were destroyed when the museum burnt in the 1990s; nevertheless there are many bones in France (Muséum national d'Histoire naturelle; in Lyon and other places), Austria in Vienna, Oslo and scattered in numerous other museums across the world, as partly listed and used by Hansford and Turvey. Most such bones were from swamps and not the dunes where the eggshell is recovered from.

We have now changed the word “scarce” to “less common” at Line 87 to clarify that, in comparison to eggshell, skeletal fossils are less abundant.

(36) Lines 98-99 - with all previously dated specimens is incorrect. Only with some dated specimens - there are sites with aepyornithid bones that are dated up to 20kyr, for example your ref 38 lists multiple sites in the late Pleistocene with aepyornithids which are thus much older than the eggshell dates. What is meant by coincide? I am not aware that bones are found with the eggshell. Yes, there are swamp deposits that have bones in them in the same southern region for example but if there are sites with both bones and shell in them then these should be detailed or at least the references listed. But I think what is meant that in the same general region there are dune deposits and swamp deposits of similar age with elephant bird remains.

The reviewer is correct: there are older dated skeletal specimens, so we have now changed “all” to “most...from these areas” at Lines 99. However, these bones have not been directly dated but rather are found in deposits that have been dated using other fossils. By “coincide” we do mean that eggshell is found in the same general area as skeletal deposits. We have changed the word “coincide” at Line 99 to “are found nearby” and have cross referenced Figure 1 where it can be seen that skeletons have been found near areas where eggshell has been found (though not necessarily at the same sites). We have also updated Figure 1 to show more localities where skeletal and eggshell material has been found, and details about these sites can be found in a new supplementary table, Table S1 which we have cross-referenced at Lines 99 and 100. Table 1 shows that Pleistocene-aged sites are still far less common than Holocene-aged sites.

(37) Line 100 – regarding ‘morphologically’ - think what is meant is ‘from skeletal material sourced from the same geographic areas’

The reviewer is correct; for clarity, we have now rephrased the sentence at Line 101-102 as per the reviewer’s suggestion.

(38) Line 102 – regarding becoming extinct soon after suggest rewrite as ‘elephant birds extinction dates more recently’ [says what you mean and lacks the imprecision of the word soon which might mean 2 years or 500 years as even the latter is very short in a geological context]

As per the reviewer’s suggestion we have removed mention of “soon thereafter” at Line 103 and replaced this phrase with “within the following few hundred years”.

(39) Lines 104-106 – I note that the ref by Gill revealed that dune sites usually revealed a bimodal distribution for moa eggshell, yet 4 species were represented. [Gill, B. J. Thickness histograms of Holocene fossil eggshell fragments indicate diversity and relative abundance of moas (Aves: Dinornithiformes) at North Island sites. *New Zealand Journal of Zoology* 49, 143-165 (2021).]

Here, we are not making any judgment about how many taxa are included within each eggshell morphotype, it is simply a description of the distribution. Genetic data from samples from across the distribution of thicknesses is what revealed how many taxa were likely represented within those distributions.

(40) Line 107 – regarding the phrase ‘represents hitherto undescribed fossils’ - think what is meant is ‘represent fossils from a new locality’; each eggshell fragment is a fossil and it is premature at this stage in the ms to conclude they relate to a unique molecular clade of aepyornithid; or is it that they represent a previously undescribed type of fossil eggshell?

While it has been mentioned once in the literature that “elephant bird” eggshell has been sighted in the far north of Madagascar, these eggshells have not been described morphologically at all until now. By this we are not implying they belong to a unique clade of aepyornithid; in fact, at this point it was unknown whether the eggshell were aepyornithid or mullerornithid. To make this clearer we have now changed the word “undescribed” to “uncharacterised” at Line 108.

(41) Lines 112-115 – I would prefer ranges with means and confidence interval as more biologically relevant. Also, what is relevant for the construction of the algorithm is the mass of the incubating bird, thus the large size of male fowl is irrelevant and likewise the large size of female ratites is irrelevant as it is the smaller males that do the incubation - hence it is their mass that should be used. Please confirm appropriate masses for taxa were used.

We have now provided means with the standard deviation for the estimated egg and body masses reported in Lines 113-117, as suggested by the reviewer. Further summary statistics can be found in Table S2. These values were recalculated and updated in the text following our re-analysis below prompted by the reviewer's comment.

We have now provided data for algorithms that used the average bird mass (both male and female; see model 1 and 2 SI 9), as well as just the male bird mass for ratites (see model 3 and 4 SI 9). Ancestral state reconstructions were performed using egg mass and body mass estimates derived from all models (Figure S11 and S12), however, we report the results derived from model 3 and 4 in the main text, as suggested by the reviewer.

It should also be noted that while the male may incubate the eggshell it is still unclear whether that is the best predictor of eggshell thickness in ratites—for instance, it may be rather that the incubation strategy more-so influences eggshell thickness; that is, how long eggs are exposed to the elements (male ostriches for instance leave the nest exposed for long periods, and thus in hot climates eggshell may be thicker relative to body mass), how many eggs are in a clutch (emus for instance lay communally, and this may affect how the incubator's weight is distributed), where on the body the incubator makes contact with the eggs (if the brood patch is near the breast less weight may be placed on the eggs), how often eggs are turned by the parent, or whether the bird's body weight is supported by its legs. It is possible that the incubator's mass is not the primary driver of eggshell thickness so without knowledge of their behaviour it is a reasonable argument to use the average bird mass as a predictor of eggshell thickness. Thus, we provide both analyses, and the results are not substantially different, however, it is interesting to note that body mass estimates are actually lower when the average (males and females) is used.

(42) Line 121-122 - Re 'We find that eggshell and skeletal specimens cluster into' I suggest edit this to say "We find that DNA haplotypes of eggshell and skeletal specimens cluster.."

We have now edited Line 124 to read “We find that mitochondrial haplotypes derived from eggshell and skeletal specimens cluster ...”, as suggested by the reviewer.

(43) Line 134 – re 'confirm the description of Mullerornis as the more gracile genus' - No - what it confirms is that the thin eggshell less than 1.5 mm is associated with the well-known genus of more gracile birds, Mullerornis. [after all the genus is based on a species Mullerornis betsilei based on bones and other taxa M. agilis, M. rudis and M. grandis and Aepyornis modestus - all based on bones-are attributed to this genus now synonymised to form one monotypic genus with the species M. modestus (Milne-Edwards & Grandidier, 1869)]

We have rephrased Line 136, as suggested by the reviewer: “confirm that the thin eggshell morphotype is associated with the gracile genus Mullerornis”.

(44) Line 141 – re families here - But for example the Howard and Moore Checklist does not have these at family level, so say who does, because it must be a quite recent change even by Clements... what is the equivalent genetic distance between the three moa families (Megalapterygidae, Dinornithidae and Emeidae)? that might be a better comparison given they have some diversity.

These checklists cite Christidis and Boles (2008), which treat emus and cassowaries both as subfamilies with the family Casuariidae. However, Christidis and Boles (2008) cite multiple

studies that retain them as separate families. Their reasoning for considering them within one family is that “this action is supported by DNA-DNA hybridisation (Sibley and Ahlquist 1981), and 12S RNA sequence (Cooper et al. 1992). The latter study showed that the divergence between the two is no greater than that between the species of kiwi *Apteryx*.” However, this study only compares a very small section of 12S RNA whereas *COI* and whole mitochondrial data shows a much more significant difference between emus and cassowaries than among kiwis (see Figure 2a)—and even more so than between moa families. The equivalent genetic distance between the three moa families is:

Emeidae – Dinornithidae: 5%

Emeidae – Megalapterygidae: 5.8%

Dinornithidae – Megalapterygidae: 3.9%

The distance between emu and cassowary is over double that. These values are reported in Table S13 and summarised in Figure S2. We believe there is a strong argument for considering emu and cassowary in separate families.

As per the reviewer’s suggestion we have now provided a reference for the family Dromiidae at Line 144. Dromaiidae is recognised as family in the Integrated Taxonomic Information System (ITIS), which cites several sources including: Gill, F., and Donsker, D., eds. 2015. IOC World Bird List (v 5.4). We have also mentioned at Line 140-141 that the genetic distance between *Aepyornis* and *Mullerornis* is “more than double the average genetic distance between moa families”), as suggested by the reviewer.

(45) Line 142 – re ‘into a new monogeneric family..’ - the family was established by Lambertson in 1934, so it is not new, rather you find support for the original hypothesis.

We have replaced the word “new” at Line 145 with “different” to clarify that they should be moved to a different family but that the family has already been named by Lambertson in 1934.

(46) Line 155 – Figure S4 - I think you mean Figure S3, there is nothing indicated in S4 or S5 to support this stmt.

Figure S5 (which was Figure S4) refers to the alignment of the XCA-2 sequences and Figure S6e-f show the annotated raw MS/MS for position 123 of XCA-2 for *Aepyornis* and *Mullerornis*. Rather, Figure S4 (what was Figure S3) refers to the XCA-1 alignment. We have not altered the reference to Figure S5 and S6 at Line 158.

(46) Line 183 – Table S9. you mean Table S19; however, this table does not give summary data for carbon nitrogen and oxygen and such would be useful rather than just the results of pairwise comparisons. Perhaps put key exemplary mean and range values here in the text.

The reviewer is correct, we have now changed the reference from “Table S9” to “Table S21” (formerly Table S19) at Line 187 (which is a summary of the pairwise comparisons, where the reported p-value can be found). However, we have not described the mean and range values here because these are shown in Figure 4b and would be redundant and too numerous—rather, the summary statistics are presented in Table S20, which we have also now cross-referenced at Line 187.

(47) Line 186 – says significantly more negative – there may be statistical significance but there is a very wide overlap as for all the data the means are -26.8 (-31.2 to -23.9) vs -25.96 (-29.4 to -21.4).

Here we are referring to statistical significance; we have now included the word “statistically” at Line 189 for clarity. We have now provided included asterisks on Figure 4b to show significant differences.

(48) Line 221 – why hav syn. *M. agilis* in brackets - the synonymy under *M. modestus* includes *M. betsilei*, *M. rudis* and *M. grandis* as well - so why mention just 1 nomen.

***M. agilis* is the only synonym mentioned because the rest were synonymised under *M. agilis* at the time the first mitochondrial DNA sequences were published for *Mullerornis*. This may cause confusion for readers searching for these sequences on GenBank which are published under *M. agilis*. However, we have now mentioned the other synonyms at Line 225, as suggested by the reviewer.**

(49) Line 235-236 – eggshell from central Madagascar remains to be tested - both eggshell and bones remain to be DNA identified.

We have now mentioned that both eggshell and bones remain to be genetically tested at Line 239-240, as suggested by the reviewer.

(50) Lines 236-237 – the sentence beginning *Prior description of numerous smaller mullerornithid..* – There are only 5 *Mullerornis* taxa named. The suggestion re sexual dimorphism and growth series is possible, but more likely, is that as these deposits are not all late Holocene and some are up to mid 20ka, that there will be temporal variation in size (related to mean temp) and this very likely explains the mean larger size of the population named as *grandis*. Note various of the moa taxa [eg *Pachyornis geranoides* (=mappini), *Euryapteryx curtus*] varied greatly (~50% reduction in size from late glacial to late Holocene; and over latitude in the Late Holocene alone; *Dinornis* populations comprise birds less than half the mean size only 200 km from the Canterbury Plains during the same late Holocene period, but at >600m altitude in montane zones) for example, see data in: Worthy, T.H., 1987. Sexual dimorphism and temporal variation in the North Island moa species *Euryapteryx curtus* (Owen) and *Pachyornis mappini* Archey. *National Museum of New Zealand Records* 3(6): 59-70.

Worthy, T.H., M. Bunce, A. Cooper, P. Scofield, 2005. *Dinornis* – an insular oddity, a taxonomic conundrum reviewed. Alcover, J. A. & Bover, P. (eds), *Proceedings of the International Symposium “Insular Vertebrate Evolution: The Palaeontological approach”* *Monographies de la Societat d’Historia Natural de les Balears* 12: 337-390. and in *Lost World of the Moa*.

The reviewer makes a very good point which we have now incorporated at Lines 242-244, citing the references mentioned: “...as well as temporal variations in size (which has been estimated to vary by over 50% between the late Glacial and late Holocene (over ca. 25 ka) in some moa species^{37,38}”.

(51) Lines 248-249 – This has a parallel in the largest individuals of the large *Dinornis robustus*, where a few big members of the larger sex stand out, see the 2005 paper just mentioned (and some of these were given taxon names, eg *maximus*, *excelsius*, and for the North Island species e.g., *hercules*) because folk have a tendency to give unwarranted value to the extremes of +20% size range around the mean in a large animal than in a small one. So, more consideration of whether these larger animals named *Vorombe* were just large females was definitely warranted before they were named.

We have now incorporated the reference mentioned by the reviewer at Line 292 and 294 to further support the point that *Vorombe* may be large females.

(52) Line 277-278 – re the stmt ‘advocate synonymising *Vorombe titan* and *Aepyornis maximus*..’ – to be precise, you mean “advocate returning the taxon *Aepyornis titan* Andrews 1894 back into synonymy with *A. maximus* Geoffroy Saint-Hilaire, 1851; and in doing so we synonymise *Vorombe* Hansford and Turvey 2020 with *Aepyornis* Geoffroy Saint-Hilaire 1851”. NOTE *Vorombe* only became available in 2020 (the correction), NOT 2018.

The reference is: Hansford J.P., S. T. Turvey, 2020. Correction to 'Unexpected diversity within the extinct elephant birds (Aves: Aepyornithidae) and a new identity for the world's largest bird'. <https://doi.org/10.1098/rsos.201358>

We have now included all the references in the reference list and cited them at Line 284, as suggested by the reviewer. We have rephrased the statement of concern (Lines 284-286) to more accurately reflect a return to the previous taxonomic hypothesis: "...we advocate tentatively synonymising *Vorombe* with *Aepyornis* and returning *titan* to synonymy with *maximus*..."

(53) Lines 282-283 - I wholly agree with the suggestion that *titan* may be the females – but it is worth drawing the analogy with *Dinornis robustus* where data show a large tail of huge forms in the large sample size available for Canterbury birds (see the 2005 paper mentioned above).

Following from Comment 51, we have now incorporated the reference mentioned by the reviewer at Line 292 and 294 to further support the point that *Vorombe* may be large females.

(54) Lines 284-285 – rewording is required. This is now dated; first, the refs 8 and 9 separated the North and the South Island populations into two species - so these taxa listed by the authors here belong *sensu stricto* only to the North Island *Dinornis* and so you should add North Island to this sentence; and it is *novaezealandiae*; so for North Island *Dinornis* these taxa became *D. novaezealandiae*, and in the South Island bones called these same 3 taxa became *D. robustus*. But, for example, in the 3-*Dinornis* system of nomenclature (in vogue 1980s-1990s) multiple taxa were already in synonymy so in the South Island, some 7 taxa including *D. torosus*, *D. maximus*, are synonyms of the single South Island *Dinornis robustus* – see the NZ checklist for correct synonymies.

We have now reworded Lines 291-293 to clarify the taxonomy, as suggested by the reviewer: "...have been consolidated into two sexually dimorphic species^{7,98}, one from the North Island (*D. novaezealandiae*) and one from the South Island (*D. robustus*). On the North Island, for instance, females ("*D. giganteus*" and "*D. novaezealandiae*") were up to 280% the mass of the males ("*D. struthoides*")." For simplicity, we just refer to the North Island *Dinornis* as an example here but specify this.

(55) Line 288 – re sexual dimorphism – suggest cite the papers on moas given above as they are the most pertinent and closest in terms of variation shown with data eg Worthy et al 2005, Worthy 1987, Worthy and Holdaway 2001.

We have now incorporated the reference mentioned by the reviewer at Line 292 and 294 to further support the point that *Vorombe* may be large females (c.f. Comment#50, 51, 53).

(56) Lines 297-299 – there is a paper on histology of aepyornithid bones..but no medullary bone was identified in the sample.

We have now cited de Ricqles et al. (2016) and Chinsamy et al. (2020) at Line 310, and mentioned the point the reviewer raises at Line 309-310. i.e., "however, thus far no medullary bone has been detected in any aepyornithid skeletal fossils thus far^{41,42}". It should be noted that Chinsamy et al. (2020) also state, "thus, owing to its ephemeral nature, the probability of finding such bone tissue among fossil birds is low". Therefore, this is a case of "absence of evidence is not evidence of absence", and we cannot necessarily conclude that because medullary bone is *not* found in *Vorombe* specimens that these were not female.

(57) Lines 310-311 – comments re diversity - Madagascar is only 1 island - moa diversity was also driven by allopatry via islands to a large degree and only 3 to 4 species ever lived together - so 4 taxa in Madagascar vs 6 in NZs SI is not greatly different and the lower diversity in Madagascar is likely explained to a great degree by other browsers being present (all the lemurs for example).

The reviewer makes a good point, and we agree with the reviewer that this statement is subjective regarding what “one might expect”; however, we think it valuable to pose the question: why are there not more species given that other lineages in Madagascar are so speciose? While it is only one island, there are other potential allopatric barriers to gene flow, such as rivers, valleys, and peaks, as well as numerous distinct climatic and ecological zones. Perhaps there is something about ratites that make them less amenable to speciation than other birds? Perhaps it is the presence of other competitors in the browsing niche (as suggested by the reviewer)? Or perhaps there are more extinct lineages of elephant birds? This statement is made more to prompt future testing of hypotheses, which could provide new insights into the dynamics of speciation.

(58) Line 313 – *the pre-Holocene skeletal record for elephant birds is scant – suggest refer to see Burney et al 2004 (A chronology for late prehistoric Madagascar) for preholocene dates, eg. Ampasamazimba (22 ka), Andrakibaikely Pond, Ankazomiranga (12-30 ka), Antsirabe (17-20 ka), and for Tsaramody (see Samonds et al 2019) =- so not really scant - were hundreds of ebird bones from Antsirabe...*

Here we were referring to the pre-Holocene fossil record in southern Madagascar—we have now clarified this by adding the phrase “from this region” at Line 325. In Burney et al. 2004, while some of the sites described *are* pre-Holocene, none are mentioned to be associated with elephant bird skeletal fossils. In fact, the only three dated elephant bird skeletal elements mentioned in Burney et al. 2004 are all dated to 1280-4496 years BP (Holocene), and the dated eggshell specimens are between 840-5210 years BP (similar to all the eggshell in this study). Ampasambazimba and Antsirabe are listed as having Aepyornithidae by Samonds et al. 2019, but these sites are in central Madagascar and no direct dates are provided. The dated skeletal specimens reported in Salmonds et al. (2019) from Christmas River are also between 9430-9535 years BP—still within the Holocene. The only pre-Holocene aepyornithid fossils (>50,000 yBP and 11,220 years BP) reported by Salmonds are eggshells. These findings have now been summarised in a new supplementary table, Table S1, which we have cross-referenced now at Line 325.

(59) Lines 331-332 - *Is it possible that some of the distinction (between the Northern and central) derives from temporal discordance as the central ones could be late glacial age, apart from the 1000 km allopatry - which has led to far great disparity in contemporary kiwi and moa populations?*

The reviewer raises a good point, however, the *A. hildebrandti* specimens genetically tested are likely contemporaneous with the other eggshells: of the dated eggshell specimens from which aDNA was able to be recovered, all date to less than 2000 years. Therefore, it is unlikely that the distinction between the northern and central clades is attributable to temporal discordance.

(60) Line 340 – *re a new species – you mean an undescribed species.*

We have changed the word “new” at Line 352 to “novel” to clarify that it may represent a species hitherto unknown to science—so not only undescribed but also different from known species.

(61) Line 342-343 – *re hildebrandti - need a reference for this - I suspect you are citing Hansford and Turvey - yet they also listed Belo-sur-Mer which is on the coast at sea level. But you need to add also Samonds et al 2019 here.*

We have now provided the recommended references at Lines 360, as suggested by the reviewer. The reviewer is correct; Hansford and Turvey (2018) do show the distribution of *A. hildebrandti* as including Belo-sur-Mer in their Figure 5b. However, their Supplementary Information (Table 1), which includes the locations for each of the skeletal elements studied only has Antsirabe and Masinandreina as localities for *A. hildebrandti* specimens. The discrepancy may arise from their Table 2 in which they synonymise *A. hildebrandti* with *A. lentus*, which is found in Belo-sur-Mer—but again, in their Supplementary Information the type specimen *A. lentus* has been assigned to

the *A. maximus* cluster. *A. lentus* was also synonymised with *A. medius* by Brodkorb (1963), and *A. medius* with *A. maximus* by Hansford and Turvey (2018). Thus, the confirmed distribution of *A. hildebrandti* remains only as central Madagascar. Tsaramody, where *A. hildebrandti* fossils were found by Samonds et al. (2019), is also in central Madagascar. Localities of fossils are now listed in Table S1.

(62) Line 410 – I do not agree with use of the word paucity - actually have been hundreds of specimens, still are hundreds in France, I rather think a 19th century philosophy of small metric differences being used to characterise taxa (as for moa) and that there have been no revisions since the 1930s except for 1 study underpin the issues. ie actual biological variation expected of a species was not applied to the taxonomy until very recently.

While the reviewer is correct about the number of specimens, most are post-cranial material, many of which are lacking diagnostic characters. At Line 430 we have included the word “diagnostic” to describe the skeletal fossils in order to clarify our meaning here.

(63) Line 423-424 – represents a novel lineage – need to add one very close to *hildebrandti*

As suggested by the reviewer we have added the phrase “likely *A. hildebrandti*” to Line 445 in the Conclusion.

(64) Line 425 – we propose a revision – this does not encapsulate that above you do synonymise *A. titan* with *A. maximus* once more and thus *Vorombe* with *Aepyornis*.

We have left this statement broad so as to include not only the re-synonymisation of *Vorombe titan* with *Aepyornis maximus*, but also other taxonomic revisions that are needed, i.e. the adoption of Mullerornithidae.

(65) Figure 2 caption - The numbers 15 and 16 as superscripts on these do not match anything obvious and are not explained in the caption.

The reviewer is correct; these numbers were intended to show which specimens had been previously sequenced by providing the reference to those papers. We have now corrected the references and included mention of this in the Figure 2 caption (Lines 473-474). We have also now mentioned this in the Figure 1 caption as well (Lines 459-460).

(66) Figure 4 caption - Opposite the panel for A are 4 sample types, yet only 3 are in panel A - is Central missing in error? The point clouds in B are not overly clear. The 5 circled ranges on the C3 half of the figure are not well enough explained and not very relatable to the habitat types in the key

Figure 4a only contains three sample types because no isotopic data was retrieved for central eggshell. We have now updated Figure 4b to make the points clearer (increased opacity and decreased diameter), and have now explained the circled ranges in the Figure 4 caption (Lines 489-490), as suggested by the reviewer.

(67) Line 605 – two authors have initial JH

The reviewer is correct; we have now referred to J Haile and J Hansford for clarity, as well as J tiller and J Sakalauskaite at Lines 666, 668, and 672.

(68) References - the references in SI 10 are in numeral form so cannot be related to the references listed in SI 11.

The reviewer is correct, we have now updated the references in SI 10 to (Author, Date) format as per the reference list in SI 11.

(69) Table 1 is confusing as the references are not well associated with the taxa. A species is described only once, though there may be extra morphological descriptions later. Strongly suggest to add the authors and date to the taxon names so they are complete taxon names; so eg *A. hildebrandti* Burckhardt 1893; Note - *Vorombe titan* was NOT described by Hansford and Turvey 2018 - the species is *Vorombe titan* (Andrews 1894) and Hansford and Turvey in 2020 (not 2018) (<https://doi.org/10.1098/rsos.201358>) erected the genus name *Vorombe* for *A. titan* Andrews 1894.

Trevor Worthy - PS if you need help with the references please ask.

We have now updated Table 1 in Figure 1 to better reflect the taxonomic references and revisions, as suggested by the reviewer. Additional references have also been included in the reference list (#93-99).

Reviewers' Comments:

Reviewer #1:

Remarks to the Author:

I thank the reviewers for their thoughtful responses to my review. I have no additional comments.

Reviewer #2:

Remarks to the Author:

Grealy and colleagues have satisfactorily addressed all of my concerns and questions. The text is now much clearer overall and more appropriately reflects the uncertainty surrounding the taxonomic status of the northern eggshells. Congratulations to the authors on an excellent study that substantially advances our understanding of elephant bird diversity. I happily recommend this study for acceptance, though I do suggest the authors consider a few last revisions:

In Figure 4, the grey ellipses are quite difficult to tell apart, especially where they overlap and the shading intensity changes. Could there be a clearer way of representing which ellipses correspond to which biome/area?

Eggshell specimens are referred to by field ID or DNA lab ID—are voucher specimens deposited at a particular institution? It would be useful to note where the remaining specimens will be physically housed (and voucher IDs, if applicable), and/or who will be responsible for them, for future replication or further research etc.

Table S3 doesn't appear to include the radiocarbon lab ID codes for each date. It would help for future reference if these codes could be included and linked to each specimen.

Reviewer #3:

Remarks to the Author:

I thank the authors for addressing all the issues previously raised. The revised manuscript is excellent and can be accepted as is.

We thank the reviewers for their constructive feedback. We have now endeavoured to take on board or address each comment below. Line numbers refer to the revised manuscript and changes have been highlighted by red text in the manuscript. We believe the reviewer's comments have helped us improve the manuscript significantly and hope the revised submission is now acceptable for publication in *Nature Communications*.

REVIEWER COMMENTS (SECOND REVISION)

Reviewer #1

1. I thank the reviewers for their thoughtful responses to my review. I have no additional comments.

We appreciate the reviewer's constructive feedback.

Reviewer #2

2. Grealy and colleagues have satisfactorily addressed all of my concerns and questions. The text is now much clearer overall and more appropriately reflects the uncertainty surrounding the taxonomic status of the northern eggshells. Congratulations to the authors on an excellent study that substantially advances our understanding of elephant bird diversity. I happily recommend this study for acceptance, though I do suggest the authors consider a few last revisions:

We appreciate the reviewer's constructive feedback and have now addressed their last concerns below.

3. In Figure 4, the grey ellipses are quite difficult to tell apart, especially where they overlap and the shading intensity changes. Could there be a clearer way of representing which ellipses correspond to which biome/area?

On closer inspection, we agree with the reviewer: we noticed that there were two ellipses representing the South-west within both C3 and CAM plants—these represented different data sources, but we have now collapsed them together for clarity. We have also deleted the ellipsis for the central-east, which is a biome not relevant to the results. Further, we have now coloured the North-west and Central ellipses to match the elephant bird samples from that region, and have also added labels directly to the ellipses to further distinguish them. Finally, the ellipses each have a distinguishing border—no border for Central, a solid border for the North-west, and a dashed border for the South-west. The figure legend has been updated. We hope this now represents clearer which ellipses correspond to which biome, as per the reviewer's request.

4. Eggshell specimens are referred to by field ID or DNA lab ID—are voucher specimens deposited at a particular institution? It would be useful to note where the remaining specimens will be physically housed (and voucher IDs, if applicable), and/or who will be responsible for them, for future replication or further research etc.

Because the eggshell was subsampled for analysis, fragments of these specimens are housed at Curtin University and The University of Colorado (Boulder), and currently they do not have voucher numbers. However, Giff Miller will be donating them to the University of Colorado Museum in 2023, after which time they will have voucher numbers. Giff Miller can be contacted

in the meantime with requests for materials. We have now included this information in the “Data availability statement” (Line 654-657).

5. Table S3 doesn't appear to include the radiocarbon lab ID codes for each date. It would help for future reference if these codes could be included and linked to each specimen.

We have now included the radiocarbon lab ID in Supplementary Data 3 (previously Table S3).

Reviewer #3

6. I thank the authors for addressing all the issues previously raised. The revised manuscript is excellent and can be accepted as is.

We appreciate the reviewer’s constructive feedback.